



# Identifying a regional aerosol baseline in the Eastern North Atlantic using collocated measurements and a mathematical algorithm to mask high submicron number concentration aerosol events

Francesca Gallo[1], Janek Uin[2], Stephen Springston[2], Jian Wang[3], Guangjie Zheng[3], Chongai Kuang[2], Robert Wood[4],
Eduardo B. Azevedo[5], Allison McComiskey[2], Fan Mei[6], Jenni Kyrouac[7], Allison C. Aiken[1]

[1]Earth and Environmental Sciences Division, Los Alamos National Laboratory, Los Alamos, NM, USA
[2]Environment and Climate Science Department, Brookhaven National Laboratory, Upton, NY, USA
[3]Center for Aerosol Science and Engineering, Department of Energy, Environmental and Chemical Engineering,
Washington University in St. Louis, St. Louis, MO, USA
[4] Department of Atmospheric Sciences, University of Washington, Seattle, USA
[5]Centre of Climate, Meteorology and Global Change (CMMG), University of Azores, Portugal
[6]Atmospheric Measurement and Data Sciences, Pacific Northwest National Laboratory, Richland, WA, USA
[7]Environmental Science Division, Argonne National Laboratory, Argonne, IL, USA

*Correspondence to*: Allison C. Aiken (aikenac@lanl.gov), Francesca Gallo (fgallo@lanl.gov)

**Abstract.** High time-resolution measurements of *in situ* aerosol and cloud properties provide the ability to study regional atmospheric processes that occur on timescales of minutes to hours. However, one limitation to this approach is that continuous measurements often include periods when the data collected are not representative of the regional aerosol. Even at remote locations, submicron aerosols are pervasive in the ambient atmosphere with many sources. Therefore, periods dominated by local aerosol should be identified before conducting subsequent analyses to understand aerosol regional processes and aerosol-cloud interactions. Here, we present a novel method to validate the identification of regional baseline aerosol data by applying a mathematical algorithm to the data collected at the U.S. Department of Energy's (DOE) Atmospheric Radiation Measurement (ARM) User Facility in the Eastern North Atlantic (ENA). The ENA Central facility (C1) includes an Aerosol Observing System (AOS) for the measurement of aerosol physical, optical, and chemical properties at time resolutions from seconds to minutes. A second temporary Supplementary facility (S1), located ~0.75 km from C1, was deployed for ~1 year during the Aerosol and Cloud Experiments (ACE-ENA) campaign in 2017.

First, we investigate the local aerosol at both locations. We associate periods of high submicron number concentration ($N_{tot}$) in the fine mode Condensation Particle Counter (CPC) and size distributions from the Ultra-High Sensitivity Aerosol Spectrometer (UHSAS) as a function of wind direction using a meteorology sensor with local sources. Elevated concentrations of Aitken mode (< 100 nm diameter) particles were observed in correspondence with the wind directions associated with airport operations. At ENA, the Graciosa airport and its associated activities were found to be the main sources of high concentration aerosol events at ENA ,causing peaks in one-minute $N_{tot}$ that exceeded 8,000 cm$^{-3}$ and 10,000 cm$^{-3}$ at C1, in summer and winter, respectively, and 5,000 cm$^{-3}$ at S1 in summer. Periods with high $N_{tot}$ not associated with these wind directions were also observed. As a result, the diverse local sources at ENA yielded a poor relationship between $N_{tot}$ measurements collected at C1 and S1 (R$^2$ = 0.03 with a slope = 0.05 ± 0.001). As a first approach to mask these events, the time periods when the wind direction was associated with the airport operations (west to northwest and southeast to south at C1 and east to south at S1) were applied. The meteorological masks removed 38.9% of the data at C1 and 43.4% at S1, and they did not significantly improve the relationship between the two sites (R$^2$ = 0.18 with a slope = 0.06 ± 0.001).

Due to the complexity of high $N_{tot}$ events observed at ENA, we develop and validate a mathematical ENA Aerosol Mask (ENA-AM) to identify high $N_{tot}$ events using one-minute resolution data from the AOS CPC at C1 and S1. After its




parametrization and application, ENA-AM generated a high correlation between $N_{tot}$ in the summer at C1 and S1 ($R^2 = 0.87$ with a slope = $0.84 \pm 0.001$). We identified the regional baseline at ENA to be $428 \pm 228$ cm$^{-3}$ in the summer and $346 \pm 223$ cm$^{-3}$ in the winter. Lastly, we compared masked measurements from the AOS with the ARM Aerial Facility (AAF) during flights over C1 in the summer to understand submicron aerosol vertical mixing over C1. The high correlation ($R^2 = 0.71$ with

a slope of $1.04 \pm 0.01$) observed between C1 and the AAF $N_{tot}$ collected within an area of 10 km surrounding ENA and at altitudes < 500 m indicated that the submicron aerosol at ENA were well mixed within the first 500 m of the marine boundary layer during the month of July during ACE-ENA. Our novel method for determining a regional aerosol baseline at ENA can be applied to other time periods and at other locations with validation by a secondary site or additional collocated measurements.

## 10  1 Introduction

### 1.1 Aerosol and cloud interactions in the Eastern North Atlantic

Ambient aerosols interact with clouds by acting as cloud condensation nuclei and affecting cloud radiative properties, with significant implications for global climate change (Anderson et al., 2003; IPCC, 2014). Currently, climate forcing associated with aerosol–cloud interactions represents one of the largest uncertainties in the climate system (Carslaw et al., 2013) and in

future climate projections (Simpkins, 2018). Compounding the effect on climate, regions dominated by clean atmospheric conditions, such as those observed in marine environments with low-lying clouds, are the most susceptible to aerosol perturbations (Rosenfeld et al., 2014). Recently, increases in larger longer-lasting cloud cover and cooling have been correlated with enhanced concentrations of aerosols in ultraclean regimes (Goren and Rosenfeld, 2015).

The Eastern North Atlantic (ENA) Ocean is a remote region characterized by a clean marine environment and persistent subtropical marine boundary layer (MBL) clouds (Wood et al., 2015). Throughout the year, transported air masses from North and Central America, Europe, the Arctic, and North Africa (O'Dowd and Smith, 1993; Hamilton et al., 2014; Logan et al., 2014) periodically impact ENA, leading to perturbations in aerosol properties and cloud condensation nuclei concentrations. As a result, ENA is one of the regions in the world with the strongest aerosol indirect forcing and, as a result, has one of the

highest associated uncertainties in terms of the aerosol impact on cloud formation, albedo, and lifetime (Carslaw et al., 2013).In the past few decades, major efforts have been focused on improving the knowledge of atmospheric processes in the ENA region. Since 1991, several campaigns including the Atlantic Stratocumulus Transition Experiment (ASTEX) (Albrecht et al., 1995), the North Atlantic Regional Experiment (NARE) field mission (Penkett et al., 1998), the International Consortium for Atmospheric Research on Transport and Transformation (ICARTT) (Fehsenfeld et al., 2006) and the BORTAS campaign

(Parrington et al., 2012) were conducted in North Atlantic studying cloud structure and long-range transport patterns over the region.

### 1.2 Ground-based aerosol measurements in the Eastern North Atlantic

Starting in 2009, the U.S. Department of Energy's (DOE) Atmospheric Radiation Measurement (ARM) User Facility has deployed campaigns at ENA to improve comprehensive long-term measurements of marine boundary layer aerosol and low

clouds in high latitude marine environments. In 2009, the 21-month field campaign (from April 2009 until December 2010), "Clouds, Aerosol and Precipitation in the Marine Boundary Layer" (CAP-MBL) on Graciosa Island (Azores Archipelago), provided the most extensive characterization of MBL clouds in North Atlantic (Rémillard et al., 2012; Rémillard and Tselioudis, 2015). The observations collected during the 21 months of the deployment also highlighted a strong synoptic meteorological variability associated to seasonal variations of aerosol properties (Logan et al., 2014; Wood et al., 2015;

Pennypacker and Wood, 2017; Wood et al., 2017).





Following the outstanding uncertainties identified during CAP-MBL and to continue the research on aerosol–cloud–precipitation interactions on marine stratocumulus clouds, in 2013, ARM established a fixed site, known as the ENA ARM Facility (Mather and Voyles, 2013; Dong et al., 2014; Logan et al., 2014; Feingold and McComiskey, 2016). The ENA fixed

site is located on the north side of Graciosa Island, which is the northernmost island within the central group of islands in the Azores. Graciosa is the second smallest in size with an area of ~61 km$^2$ and is one of the least populated islands within the Azores archipelago, with a population of less than 5,000 people. These features make Graciosa Island well-suited for collecting measurements representative of the open ocean from an inhabited island with power and infrastructure.

The ENA Central Facility (C1) is equipped with an Aerosol Observing System (AOS). The AOS provides a unique dataset of high temporal resolution measurements of *in situ* aerosol optical, physical, and chemical properties and their associated meteorological parameters (Uin et al., 2019). Most recently, motivated by the need of a characterization of the horizontal variability and the vertical structure of aerosol and clouds over ENA, ARM deployed the Aerosol and Cloud Experiments in the Eastern North Atlantic (ACE-ENA) field campaign (Wang et al., 2019a). In July 2017 during ACE-ENA, ARM established

a temporary Supplementary facility (S1), approximately 0.75 km from the central ENA site (C1), to understand the regional representativeness of the AOS data at the ground-level. A subset of AOS instruments was deployed for a period of approximately one year to identify the local impacts at C1 and to add additional constraints for the development of algorithms to mask local aerosol influences. During two Intensive Operating Periods (IOPs), in June-July 2017 and January-February 2018, the ARM Aerial Facility (AAF) Gulfstream-159 (G-1) research aircraft flew over ENA providing high quality

measurements of the marine boundary layer and lower free troposphere (FT) structure, as well as the vertical distribution and horizontal variability of low clouds and aerosol over ENA (Wang et al., 2016; Wang et al., 2019a). We use the AAF and S1 data to constrain periods of time when the ENA AOS data was regionally representative of aerosol concentrations at the ground level and when they represented aerosol concentrations that were well-mixed within the boundary layer.

### 1.3 Masking local aerosol sources

The impact of local sources on aerosol and trace gas measurements is a common issue for continuous ambient datasets (Drewnick et al., 2012). Even at remote sites such as ENA, local sources can be pervasive and  unavoidable. At ENA, the location for C1 was selected by ARM to minimize local aerosol and trace gas sources since they can interfere with regional and large-scale atmospheric aerosol processes. However, competing needs of instruments, logistics, and operations (e.g. requirement of large flat surface areas for the radars, power and infrastructure to operate the facility, etc.) constrained the site

selection. As a consequence, episodes of local aerosols are sampled by the AOS and can be observed in the high time-resolution data. Thus, we identify all known local sources and develop a mask to isolate the regionally representative data.

One method to estimate the regionally representative concentrations at sites affected by local aerosol is with meteorological filters (Giostra et al., 2011; Gao et al., 2019; Wang et al., 2019b). This approach masks all data related to air masses coming

from wind directions associated with sources. However, meteorological filters rely upon accurate knowledge of the local sources and the availability of high quality meteorological data (Giostra et al., 2011). This method has limited use at locations where local sources originate from a wide range of wind directions, vary with time, such as seasonal sources, and at locations with complex meteorology, terrain, and high wind speeds.

With high time-resolution continuous data, it is possible to implement post data processing methods based using statistics to identify and mask high concentration aerosol events without removing a large fraction of the data or relying on observational data to identify nearby sources. Mathematical algorithms that evaluate the statistically different behavior of adjacent data



points have been shown to be effective for masking real-time atmospheric data affected by local events in clean environments (Giostra et al., 2011; McNabola et al., 2011; Drewnick et al., 2012; Hagler et al., 2012; Ruckstuhl et al., 2012; Brantley et al., 2014; Wang et al., 2015). The challenge, however, is to identify and mask the time periods impacted by local aerosol sources without masking the regionally representative data that may include periods of long-range transport or other sources with high

aerosol number concentrations. Hence, for the successful application of mathematical algorithms, it is important to know how local sources impact the measurements, especially in terms of the signal change and duration of the events, to appropriately configure the algorithm (El Yazidi et al., 2018; Wang et al., 2019b). In this context, collocated and/or additional nearby aerosol and trace gas data are useful to understand the origins and pervasiveness of local aerosol and to validate the application of different masking algorithms.

The first aerosol filter applied to ENA AOS data by Zheng et al. (2018) was used to study seasonal aerosol–cloud interactions. The authors used AOS CPC data to mask time periods when the first derivative of the submicron aerosol number concentration exceeded 60 particles $cm^{-3}$ $s^{-1}$. With this method, < 20% of data was masked within each one-hour averaging interval. Other potential methods, that require further development and validation, include the application of machine learning. The External

Data Center (XDC) identified periods in ENA AOS data that were impacted by local combustion sources due to planes and runway operations at the Graciosa airport over a five day time period during the winter (Mitchell et al., 2017). Future efforts to develop and apply this code at ENA should be investigated but were beyond the scope of the work presented here.

We present data from two facilities at ENA, C1 and S1, during ACE–ENA to identify the local aerosol sources at ENA and to

determine their influence on the AOS data. Submicron aerosol concentrations, size distributions, and meteorological data collected are presented. We develop a new aerosol mask at ENA using AOS data to identify periods of short-duration high concentration submicron particle events. Our mathematical algorithm and the determination of a regional baseline for submicron aerosol is validated using the data from C1 and S1. After determining the regional baseline, we compare AOS masked data with the AAF data collected during ACE-ENA flights over C1 to understand the vertical distribution of aerosol

at ENA.

## 2 Measurements

### 2.1 ENA Central Facility (C1) and aerosol Supplementary site (S1)

The ENA Central facility (C1) is located on Graciosa Island within the Azores Archipelago at 39° 5' 28" N, 28° 1' 36" W. C1 is located on the northern part of the Island as the area is flat, has access to local power, and is mostly unpopulated (Fig. 1).

High temporal-resolution measurements (seconds to minutes) of aerosol properties at C1 are made with the ENA AOS (McComiskey and Ferrare, 2016; Uin et al., 2019). The AOS at ENA C1 includes instruments for measuring aerosol optical, physical, and chemical properties, trace gases, and meteorological parameters. The AOS is comprised of one container that samples aerosols using instrumentations connected to a central inlet located approximately 10 m above ground level (Bullard et al., 2017; Uin et al., 2019).


The Aerosol Supplementary site (S1) was deployed at 39° 5' 43" N, 28° 02' 02" W, ~ 0.75 km from C1 (Fig. 1), in July of 2017. S1 was sited within 1 km of C1 to maintain the relevance of S1 data to the AMF measurements at C1. S1 was located at ~0.2 km from the shore (closer than C1) at ~50 m a.s.l. Data was collected at S1 until the site was decommissioned in April of 2018 after the conclusion of ACE-ENA.






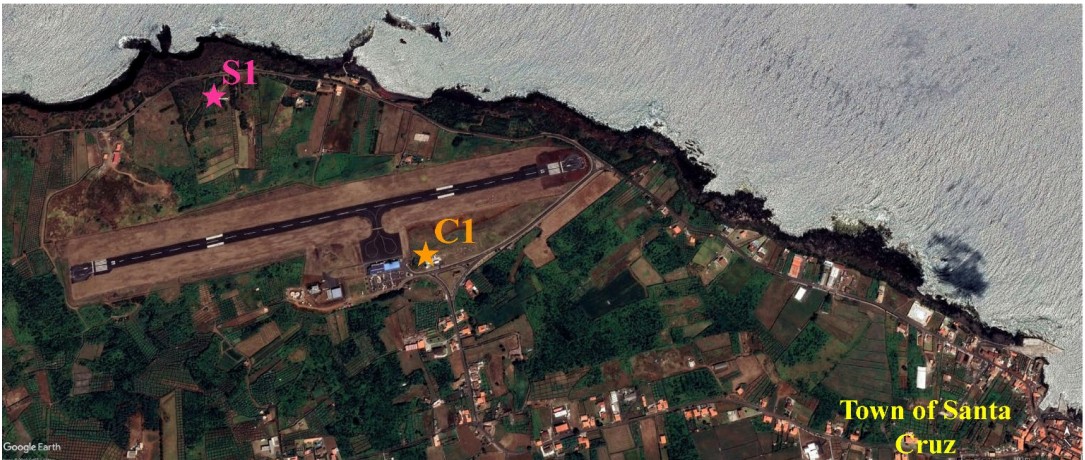

**Figure 1.** Satellite image of ENA C1 and S1 on Graciosa Island, Azores, Portugal (© Google Earth).

Three instruments, duplicate models of those used within the AOS at C1, were deployed at S1. Two aerosol instruments were

selected for their ability to measure submicron aerosol concentrations in high time resolution. The third instrument was included to associate the measurements with meteorological parameters as is done in the AOS. The aerosol instruments were powered and located inside a converted garage in an unoccupied house, with the computer for data acquisition. The meteorology sensor was mounted above the inlet at ~3 m above the roofline. Measurements were designed to duplicate those made within the AOS as best possible without the use of an AOS inlet at S1. Prior to the deployment at S1, the instruments

were calibrated at C1 alongside the AOS instruments. Ambient data from the three instruments were compared over a period of one week at S1. The S1 inlet flow rate was optimized to minimize submicron particle loss (Bullard et al., 2017).

Briefly, the fine mode Condensation Particle Counter (CPC) (TSI, Inc., Shoreview, MN, USA; Model 3772) measures the submicron number concentration ($N_{tot}$) of aerosols from ~ 7 nm to 1 μm in particle diameter ($D_p$). Particles are grown by

condensing butanol vapour onto the particles before they are optically counted by illuminating them with a laser beam to count the number of light pulses that are scattered (Kuang, 2016). The Ultra-High Sensitivity Aerosol Spectrometer (UHSAS) (Droplet Measurement Technologies, Inc., Longmont, CO, USA) is an optically scattering, laser-based aerosol spectrometer for sizing particles from ~ 60 to 1000 nm $D_p$. Aerosols scatter the laser light as a function of their optical $D_p$. The UHSAS detection efficiency is ~100% for particles > 100 nm and for concentrations < 3,000 cm$^{-3}$ (Cai et al., 2008). Concentration

measurement errors occur for smaller particles that have low scattered light intensities and during periods of higher $N_{tot}$ due to particle coincidence. Sizing of spherical and irregular particles by the UHSAS are within 10% of the mobility diameters measured by the Scanning Mobility Particle Sizer (SMPS) for particles with $D_p$ > 70 nm (Cai et al., 2008). Therefore, in this study, we use the UHSAS submicron data for particles > 70 nm (Uin, 2016). Since submicron data was collected at S1 and compared with the submicron data collected at C1, we make no inferences on supermicron particles. The meteorology sensor

(Met) (Vaisala, Finland; WXT520), provides ambient air temperature, relativity humidity, atmospheric pressure, wind speed and direction relative to true North, and precipitation data (rain amount, duration and intensity) (Kyrouac, 2016).

## 2.2 ARM Aerial facility (AAF)

The ARM Aerial Facility (AAF) Gulfstream-159 (G-1) research aircraft flew from Terceira Island (~90 km from the ENA C1 site) during two IOPs in early summer 2017 (June to July) and winter 2018 (January to February). Flight patterns included

spirals, to obtain vertical profiles of aerosol and clouds, and ascendant and descendent legs at multiple altitudes, to provide





characterization of the boundary layer and lower free troposphere structure. Data were collected up to an altitude of ~4,970 meters.

$N_{tot}$ collected by the AAF with the CPC (TSI, Inc., Shoreview, MN, USA; Model 3772) during the summer were compared to
CPC data collected at C1 and S1. The CPC was installed behind an isokinetic inlet to minimize particle loss in aircraft sampling and was operated on the AAF G-1 (Schmid et al., 2014). During the first ACE-ENA IOP there were 20 Research Flights (RF). We analyzed data from the seven flights that collected data over C1 at altitudes between 54 and 500 meters.

## 3 Data Analysis

### 3.1 C1 and S1 intercomparison

We present and evaluate different strategies to identify periods when the AOS data are impacted by high submicron aerosol concentrations and associate them with nearby potential aerosol sources. The impact of local aerosol sources at ENA C1 are evaluated by comparing data collected at C1 and S1. We analyzed two one-month time periods that represent two seasons: summer (7/22/17 – 8/20/17) and winter (12/01/17 – 12/30/17).

Measurements from the USHAS and CPC are combined to describe the submicron aerosol size distribution by dividing the data into three optical size modes. Zheng et al. (2018) used lognormal fitting of the submicron aerosol size distributions from the UHSAS to define three modes to study aerosol–cloud interactions at ENA. The lognormal fittings gave three parameters: mode diameter, mode number concentration, mode σ (Table 2 in Zheng et al. (2018)). Number concentrations ($N$) of the fitted modes were classified by the mode diameter as: 1) $N_{At}$, number concentration of Aitken (At) mode aerosol ($D_p \leq 100$ nm), 2)
$N_{Ac}$, number concentration of Accumulation (Ac) mode aerosol ($D_p = 100 - 300$ nm), and 3) $N_{La}$, number concentration of Large accumulation (La) mode aerosol ($D_p = 300$ 1000 nm). The $N_{Ac}$ and $N_{LA}$ mode number concentrations reported here are directly measured by the UHSAS. Since there is not a direct measurement of the full range of At mode particles, $N_{At}$ is determined by combining the measurements from the CPC and the UHSAS. $N_{At}$ is calculated as the difference between the $N_{tot}$, as measured by the CPC, and the sum of the UHSAS number concentrations from the two larger modes: $N_{At} = N_{tot} - (N_{Ac}$
$+ N_{LA})$. All $D_p$ referenced in the text refer to aerosol optical diameter unless they are stated as otherwise.

One way to determine statistical outliers in the data is by comparing the difference between the median and the mean. Time periods when the median and mean $N_{tot}$ differ significantly are used to indicate periods when the data is affected by outlying events, such as high number concentration aerosol events. Median values represent the midpoints in the data, which are
minimally affected by outlying events. Mean values describe the central tendency of the data and are affected by outlying events. As such, comparison between the two values provides information about the variability within the overall dataset. Erroneous data and their QA/QC flags (e.g. negative values and -9999) have been removed prior to the analysis presented here. Significant deviations between the mean and median concentrations, where the mean is biased high, are used to indicate when aerosol $N_{tot}$ have a statistically relevant higher variability due to the presence of high concentration aerosol events.

### 3.2 ENA Aerosol Mask (ENA-AM)

ENA Aerosol Mask (ENA-AM) is a standard deviation algorithm that was parameterized for the ENA $N_{tot}$ data collected by the CPC. Application of the algorithm requires the statistical differences between adjacent data points to distinguish periods of short-duration high aerosol number concentrations from the baseline measurements. The time resolution of the data has to be shorter than the typical time period of the high concentration events. The variation within the clean baseline periods also
has to be smaller than the variation of $N_{tot}$ during the high concentration events. Therefore, the algorithm works best with high



time-resolution data, as is collected by the AOS at time intervals on the order of seconds to minutes, and for identifying local sources that have high temporal variability. An additional requirement is that at least half of the total data points have to be representative of the baseline conditions otherwise the algorithm is not able to identify the high $N_{tot}$ events properly. The one-minute $N_{tot}$ data collected at C1 and S1 fulfil these requirements, and we, therefore, developed ENA-AM as described below

using two one-month periods of data collected at ENA (Gallo and Aiken, 2020).

We determined the standard deviation of the data below the median ($\sigma_b$) of $N_{tot}$ for each of the two one-month periods. Any data point that differs by more than $\alpha$ times the $\sigma_b$ from the preceding data points is identified and masked as a high concentration aerosol event. The retained data points are defined as the baseline. The variable $\alpha$ is used to set the threshold,

and its value is defined as a function of the specific dataset and time series variability. An alternative parameterization would be to use the standard deviation of the data that were between the first and the third quartiles. While this alternative was not explored here, we expect that it would yield similar results.

Whenever a data point is identified above the threshold, the next point in the time series is evaluated using a random walk

method (threshold = ($\sigma_b$ + sqrt (n)) * $\alpha$), where n is the number of data points since the last data point that was within the standard variability. In this way, the threshold is slightly increased to account for normal temporal development of the baseline. If the density of the high concentration particle events is high, the algorithm is not able to identify the baseline variability properly. In such cases, $\alpha$ should be set to a lower value, and the random walk method threshold might be better substituted with a two-point thresholding method. With two-point thresholding, the two data points after each masked point are considered

to be part of the event. Thus, the value of $\alpha$ and the thresholding method are dependent on the time series variability as is the selection of the time period over which to apply the algorithm. Selection of both must be optimized for the specific dataset. We tested four different parametrizations of the algorithm, which included two $\alpha$ values and the two thresholding methods as well as different time lengths. Table 1 presents the combination of $\alpha$ values and thresholding methods used. A sensitivity analysis was conducted to determine the best parametrization of the algorithm for $N_{tot}$ measurements at ENA. After the

parameters for ENA-AM were determined, we compare masked C1 with AAF $N_{tot}$ measurements.

**Table 1.** Standard deviation algorithm input parameters tested at C1 and S1 in the summer.

|  | Random Walk (RW) Threshold | Two-Point (TP) Threshold |
|---|---|---|
| $\alpha = 1$ | $\alpha$1-RW | $\alpha$1-TP |
| $\alpha = 3$ | $\alpha$3-RW | $\alpha$3-TP |

## 4 Results and discussion

### 4.1 Wind direction and speed

The percentage of time that the wind was sampled from a particular direction during the summer and winter are reported in Table 2, as a function of the cardinal and intercardinal directions. During the summer and winter at C1 and S1, the winds dominantly come from the southwest and south. Minimal time periods were sampled when the wind was coming from the north and northwest.

During the summer, C1 and S1 were dominated by winds from southwest at 27.6% and 32.8% of the time, respectively. At C1, winds from the east at 14.6% and southeast at 15.0% were the next most dominant directions. At S1, winds from the east were also the next highest at 21%, while the southeast wind direction was sampled less often at 8.2%. Winds from the north and northwest were the least frequent at C1 and S1: north at 7.3% (C1) and 6.9% (S1), northwest at 6.7% (C1) and 5.6% (S1).




In the winter, the wind had an almost equally dominant contributions from the south, at 29.0% (C1) and 30.3% (S1), and southwest, at 30.9% (C1) and 30.8% (S1). At C1 the next largest wind directions sampled were from the west at 17.6% and from the southeast at 12.6%. S1 differed from C1 in that, while the next most dominant wind direction was from the southeast

5     at 19.4%, the wind from the west was significantly less at 7.4%. In contrast to the summer, both sites had a negligible contribution (≤ 12%) from the wind directions associated with the direction of the shore that equates to half of the wind rose: northwest, north, northeast, and east. While these wind directions were also not the dominant wind directions in the summer, the difference was that the winds were more equally distributed in the summer at 40.3% (C1) and 41.8% (S1) from the northwest to east.

**Table 2.** Percentage of time sampled as a function of wind direction during summer and winter at C1 and S1.

| Wind Direction | Summer | | Winter | |
|---|---|---|---|---|
| | C1 | S1 | C1 | S1 |
| N | 7.3% | 6.9% | 2.4% | 2.9% |
| NE | 11.7% | 8.0% | 2.0% | 1.7% |
| E | 14.6% | 21.3% | 2.3% | 5.6% |
| SE | 15.0% | 8.2% | 12.6% | 19.4% |
| S | 6.9% | 10.6% | 29.0% | 30.3% |
| SW | 27.6% | 32.8% | 30.9% | 30.8% |
| W | 10.3% | 6.5% | 17.6% | 7.4% |
| NW | 6.7% | 5.6% | 3.2% | 1.9% |

The frequency of wind speed sampled at ENA are shown as a function of wind direction in Fig. 2. In general, the surface wind speed was higher at C1 than at S1, independent of the season. In the summer (Fig. 2a, b), surface wind speed mean values and

15     one standard deviations were $4.7 \pm 2.3$ m s$^{-1}$ (C1) and $3.2 \pm 1.6$ m s$^{-1}$ (S1). The maximum wind speed during the summer came from the southwest for both sites: 15.2 m s$^{-1}$ (C1) and 10.0 m s$^{-1}$ (S1).

In the winter (Fig. 2c, d) the mean wind speeds at ENA was approximately double the speeds measured in the summer. The mean wind speed and standard deviations recorded were $7.3 \pm 2.5$ m s$^{-1}$ (C1) and $5.7 \pm 2.0$ m s$^{-1}$ (S1). The peak wind speed

20     measured during the winter was from the same direction as it was in the summer, from the southwest: 21.7 m s$^{-1}$ (C1), 16.6 m s$^{-1}$ (S1).





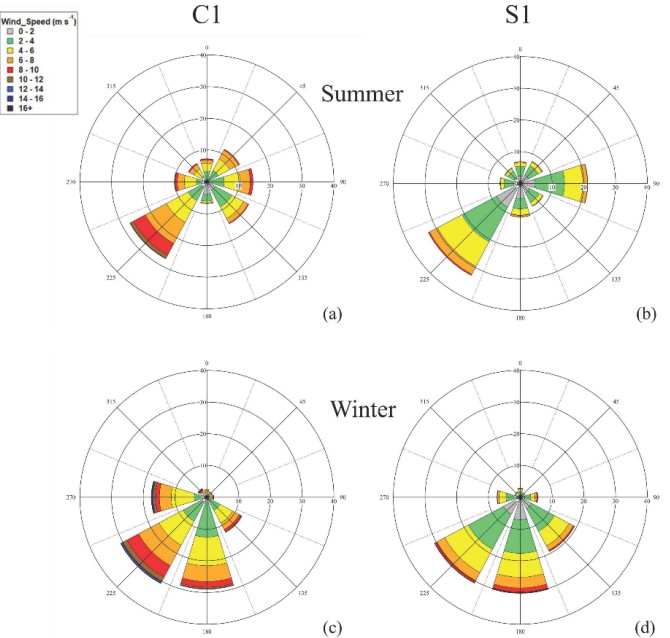

**Figure 2.** Surface wind rose plots during the summer (a, b) and winter (c, d) at C1 and S1. The length of the radial bars is the frequency of different wind speed ranges shown in colour, as a percentage of the time sampled.

The mean wind speeds observed at C1 and S1 were within 25% of each other during the summer and winter. The wind speed at C1 was higher during both seasons. S1 mean winds were 68% and 78% of C1 in the summer and winter, respectively. The observed maximum wind speed coincided with the dominant wind direction at C1 and S1 in both seasons. This observation was most evident in the summer when the wind was dominated by winds from the southwest. During the winter, there was a similar fraction of wind from the south and southwest with higher average wind speeds at both sites. Overall, C1 and S1

experienced similar dominant wind directions and mean wind speeds. We, therefore, expect C1 and S1 to exhibit similar trends in the aerosol data with the exception of the time periods when they are impacted by local aerosols that are not representative of the region. Those time periods should be influenced by the proximity, direction, size and type of the aerosol source in relation to the measurement site.

**4.2 High concentration aerosol events**

Wind directions can be used with aerosol measurements to determine aerosol sources (Zhou et al., 2016; Cirino et al., 2018). To understand the frequency and direction from which local aerosols originate at ENA, we present mean aerosol $N_{tot}$ and $N_{UHSAS}$ as a function of wind direction. $N_{tot}$ and $N_{UHSAS}$ are used to understand the directional and temporal influence of observed high aerosol concentrations at C1 and S1 and to evaluate the use of wind direction data to create and aerosol mask at ENA.

In Fig. 3, one-minute $N_{tot}$ and $N_{UHSAS}$ are averaged as a function of wind degree direction in the summer and winter. When plotted by wind degree direction, we observed $N_{tot} > 1,000$ cm$^{-3}$ at C1 and S1. Mean $N_{tot}$ for all directions in the summer were 710 cm$^{-3}$ (C1) and 490 cm$^{-3}$ (S1). $N_{UHSAS}$ mean concentrations were less than half of $N_{tot}$ during the same time periods: 342 cm$^{-3}$ (C1) and 210 cm$^{-3}$ (S1). The higher $N_{tot}$ is due to a significant fraction of aerosol below the UHSAS lower detection size limit of 70 nm since the instruments have similar upper limits for counting particles. Without the $N_{tot}$ that counts particles < 70 nm

$D_p$, the high concentration aerosol would be harder to identify by wind direction alone due to the lower variability in $N_{UHSAS}$. For this reason, we continue our analysis by wind direction focusing on $N_{tot}$.





The largest mean $N_{tot}$ plotted by wind degree direction that was observed at C1 was $\geq 3,000$ cm$^{-3}$ (Fig. 3a, c) during summer and winter when the winds were from the west to northwest, wind directions that are associated with the airport (see Supplement Information (SI) and Table S1). These directions were attributed to the utilization of the runway and the airplane

parking lot with AOS camera visual validations of aircraft. The next highest $N_{tot}$ were observed from the south to southeast at C1. $N_{tot} \geq 1,000$ cm$^{-3}$ were observed in the summer and $N_{tot} \geq 1,600$ cm$^{-3}$ in the winter. These directions are associated with the direction of the road that leads from the airport to the town of Santa Cruz (Figs. 1 and S1).

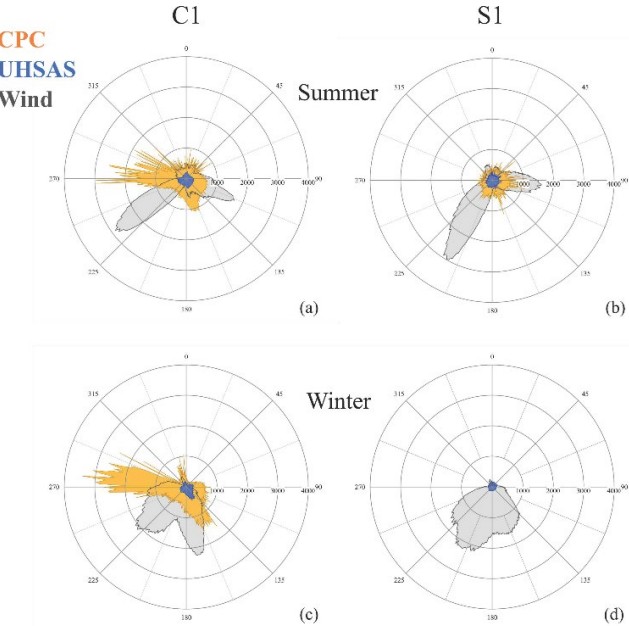

**Figure 3**. Polar graphs of the mean $N_{tot}$ and $N_{UHSAS}$ as a function of wind direction during summer (a, b) and winter (c, d) at C1 and S1. One-minute $N_{tot}$ from the CPC, in orange (data not available at S1 in the winter), and $N_{UHSAS}$ from the UHSAS, in blue, were averaged as a function of wind degree direction. The frequency of wind direction is in grey.

While mean $N_{tot}$ were lower at S1 than C1, S1 also had $N_{tot} > 1,000$ cm$^{-3}$ (Fig. 3b). The three highest $N_{tot}$ at S1 that exceeded

1,000 cm$^{-3}$ were observed from the south-southeast, east-southeast and east. The wind directions of the maxima $N_{tot}$ were associated with the airport runway, rural road, and pasture at S1. Wind directions with $N_{tot} \sim 1,000$ cm$^{-3}$, observed from the northeast, were likely due to the rural road along the shore. $N_{tot} \sim 500$ cm$^{-3}$ from the southwest were from the direction of the decommissioned landfill that still has active vents as well as the airport runway. $N_{tot}$ was not available during the winter at S1 to make a comparison with summer.

The results of the wind direction analysis indicates that the main sources of $N_{tot} \geq 1,000$ cm$^{-3}$ at C1 and S1 are most likely associated with airport activities and road traffic due to the proximity and direction of the sources. However, at ENA, other unattributed local sources, not related to airport operations could also be present that are not identified here. One example of an aerosol source that we could not verify was a potential brick production facility $\sim 1$ km to the south-southeast of C1.

Complex meteorological conditions known to exist in the region might also be responsible for high $N_{tot}$ at C1 and S1 that we were not able attribute to local sources based on wind direction.



### 4.3 Size-resolved submicron aerosol

### 4.3.1 Size distributions

Size distributions can also be used to determine the source of high aerosol concentrations since different combustion sources, fuel types, and modes of operation produce different particle sizes. For example, depending on the jet fuel composition,

aircrafts produce Aitken mode particles with $D_p < 100$ nm at number concentrations of ~ 4,000 – 30,000 cm$^{-3}$ (Moore et al., 2017) that have been observed to be even greater during take-off and landing at $\geq 40,000$ cm$^{-3}$ (Campagna et al., 2016). Aerosol from port fuel injection gasoline vehicles have been measured to be ~ 10,000 cm$^{-3}$ with mean $D_p$ between 40 to 80 nm. The $N_{tot}$ of particles emitted by diesel engines can be as high as $10^4$ cm$^{-3}$ with larger mean aerosol $D_p$ between 60 and 120 nm, larger than those emitted by gasoline engines (Harris and Maricq, 2001). In general, combustion engines produce mean aerosol

size distributions dominated by the Aitken mode. Smaller size distributions are attributed to more efficient combustion sources and shorter distances between the source and measurement site (Lighty et al., 2000).

In Fig. 4, we compare the measured size distributions from the UHSAS at C1 and S1 in the summer and winter. During the summer, the size distributions were similar in the number concentration and the mean mode $D_p$ of 150 nm at both sites. A

bimodal size distribution was evident, but the peak of the smaller mode was not able to be determined due to the UHSAS lower $D_p$ limit. Despite not being able to size particles below $D_p = 70$ nm, the combined analysis presented in Section 4.2 enabled us to conclude that similar number concentrations of particles observed at C1 and S1 occurred above and below $D_p = 70$ nm. The main difference observed between the measured size distributions at C1 and S1 in the summer was that there were 11% more particles from 80 nm to 150 nm at C1 than S1 (Fig. 4a).

In the winter, the bimodal structure was less evident at C1 and S1 (Fig. 4b). C1 had 38% more aerosol with $D_p < 150$ nm than S1. This difference was significantly larger than what was observed in the summer. The peak of the size distribution was also shifted to slightly smaller sizes than 150 nm at C1 than S1. This could have been due to the presence of more local sources with aerosols $D_p < 150$ at C1 than S1 in the winter and/or to different meteorological conditions than were observed in the

summer. This difference in season was in agreement with Fig. 3 (Sect. 4.2) where a higher difference between $N_{tot}$ was observed between C1 and S1 in the winter than in the summer.

During both summer and winter, C1 had more aerosol than S1 for $D_p < 150$ nm. This is likely due to the closer proximity of C1 to the Graciosa airport and the road to Santa Cruz. While the difference between the size distributions measured at C1 and

S1 in the summer was minimal for $D_p > 70$ nm, we expect the difference would be more evident if sizing were available for $D_p < 70$ nm. We base this statement on the comparison of $N_{tot}$ and $N_{UHSAS}$ in the previous section. For this reason, we combined the information from the two instruments to probe the Aitken mode aerosol in the next section.





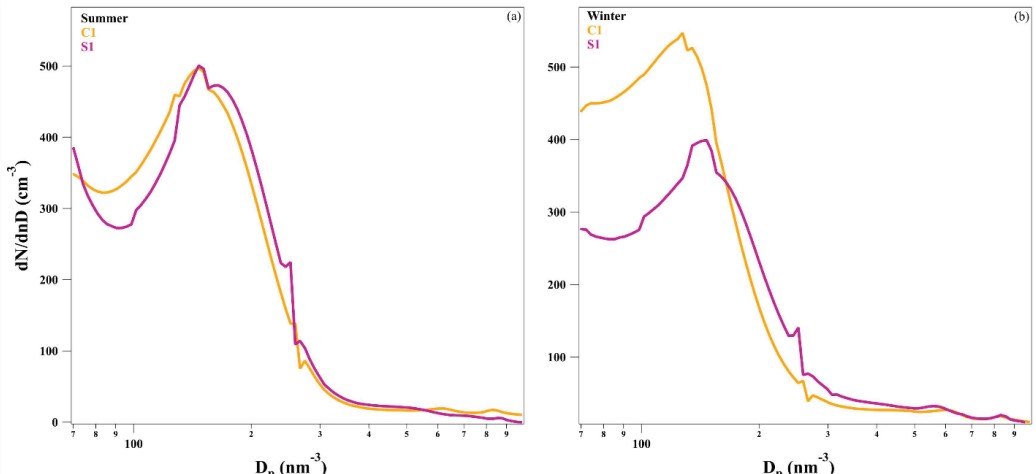

**Figure 4**. Aerosol size distributions at C1 (orange) and S1 (pink) during summer (a) and winter (b).

### 4.3.2 Submicron aerosol modes

Number concentrations from three aerosol modes that we defined in Section 3.1 are presented in Fig. 5 from C1 and S1 in the

summer and winter. The smallest mode number concentration, $N_{At}$, represents the size range most likely impacted by nearby combustion sources as discussed in the Section 4.3.1. $N_{Ac}$ is expected to include some of these particles as well, especially for the less efficient combustion sources and operational modes, such as those produced by diesel engines and wood burning sources that may or may not be significant at ENA, and are not discussed here due to their unconfirmed use on the island. The third and largest mode number concentration, $N_{LA}$, is not expected to be significantly impacted by nearby combustion aerosol

sources. However, $N_{LA}$ is presented since it includes natural aerosol sources such as sea spray (Burrows et al., 2014; Quinn et al., 2015) and secondary organic aerosol (Jimenez et al., 2009; Shrivastava et al., 2019) that can also be formed in association with combustion sources.

For the three submicron size modes analyzed at C1 and S1, $N_{At}$ had the largest median and mean number concentrations,

equating to 44% of the median and 31% of the mean for the total submicron aerosol concentrations, $N_{tot}$, when averaged from the different sites and seasons. $N_{At}$ also had the highest deviation between the mean and median of the three size modes during the summer and winter. $N_{At}$ at C1 were relatively constant at 245 cm$^{-3}$ in the summer and 258 cm$^{-3}$ in the winter. $N_{At}$ at S1 was 78% of C1 with 190 cm$^{-3}$ in the summer. While median $N_{At}$ were relatively constant for the data shown here at both seasons and sites, mean $N_{At}$ varied with site and season. Mean $N_{At}$ were 540 cm$^{-3}$ (C1) and 330 cm$^{-3}$ (S1) in the summer (Fig. 5a). In

the winter at C1, the mean $N_{At}$ was 800 cm$^{-3}$, which was 48% higher than what was observed in the summer (Fig. 5b).

The higher observed mean $N_{At}$ at C1 during the winter indicated that the influence of nearby aerosol sources was likely to be larger in the winter than in the summer. This result is supported by the earlier results from $N_{tot}$ (Section 4.2: Fig 3a, c) and the submicron size distributions (Section 4.3.1: Fig. 4b). The reason for the higher fraction of $N_{At}$ observed in the winter at C1

could have been due to additional seasonal sources that were not attributed here, such as the burning of wood or other fuels to heat homes, etc. Different meteorological conditions experienced in the winter versus the summer could also have contributed to the seasonal differences. For example, higher $N_{At}$ from the known sources discussed in Section 4.3.1 might also be different winter meteorological conditions, e.g. lower boundary height, higher wind speeds, etc.





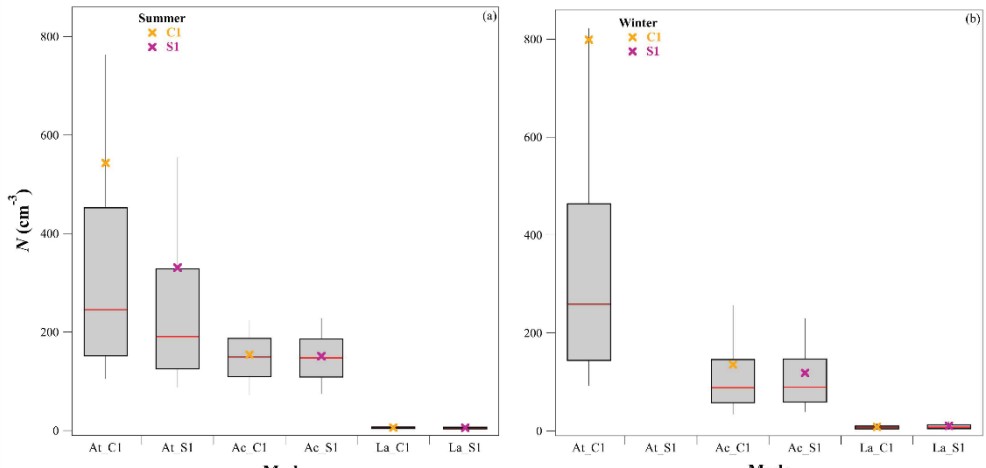

**Figure 5**. Box and whisker plot of At, Ac and La mode aerosol number concentrations at C1 (orange) and S1 (pink) in the (a) summer and (b) winter. Mean (x) and median (red line). Box bottom at 25%, box top at 75%, whisker bottom at 10%, and whisker top at 90%. No At mode data was available at S1 during the winter.

Mean and median $N_{Ac}$ were lower than $N_{At}$ during summer and winter at C1 and S1, yet still represented a significant fraction of $N_{tot}$. In the summer, C1 and S1 $N_{Ac}$ had similar mean and median values, indicating low variability in the data. Mean $N_{Ac}$ observed were 160 cm$^{-3}$ (C1) and 151 cm$^{-3}$ (S1). Median $N_{Ac}$ were 156 cm$^{-3}$ (C1) and 147 cm$^{-3}$ (S1). The similar values between the mean and median $N_{Ac}$ at both sites indicated that the mode was not largely affected by high concentration aerosol events. In the winter, mean $N_{Ac}$ were 13% (C1) and 22% (S1) lower than mean $N_{Ac}$ in the summer. Mean $N_{Ac}$ were 140 cm$^{-3}$ (C1) and 118 cm$^{-3}$ (S1). Median $N_{Ac}$ were 91 cm$^{-3}$ (C1) and 89 cm$^{-3}$ (S1). Overall, $N_{Ac}$ at C1 and S1 were more similar than $N_{At}$ in summer and winter. However, there was a higher variability between the mean and median $N_{Ac}$ observed during the winter that was not observed in the summer.

$N_{LA}$ did not represent a significant fraction to $N_{tot}$ at ENA for the data presented here. Mean $N_{LA}$ during the summer were 6 cm$^{-3}$ at C1 and S1. Similar $N_{LA}$ were observed in the winter at 8 cm$^{-3}$ at C1 and 10 cm$^{-3}$ at S1. While $N_{LA}$ is important in regard to mass concentrations, scattering properties, and cloud condensation nuclei, all properties measured by the AOS (Uin et al., 2019), $N_{LA}$ are not generally attributed to local combustion aerosol sources, which was the focus here. Contributions and impacts to $N_{LA}$ due to sea spray aerosol were beyond the scope of this work, yet were not considered to be a large contribution at C1 or S1 based on the low $N_{LA}$ observed here.

### 4.3.3 Variability with wind direction

Mean $N_{At}$ had the highest variability of all modes as discussed above. $N_{At}$ and $N_{Ac}$ had a larger observed variability in the winter than in the summer. For these reasons we evaluate the dominant submicron size modes as a function of wind direction to assess the variability in association with the direction of the known potential local aerosol sources identified in Section S1 in the Supplement. $N_{At}$ and $N_{Ac}$ are plotted in Fig. 6 at C1 and S1 during the summer and winter.


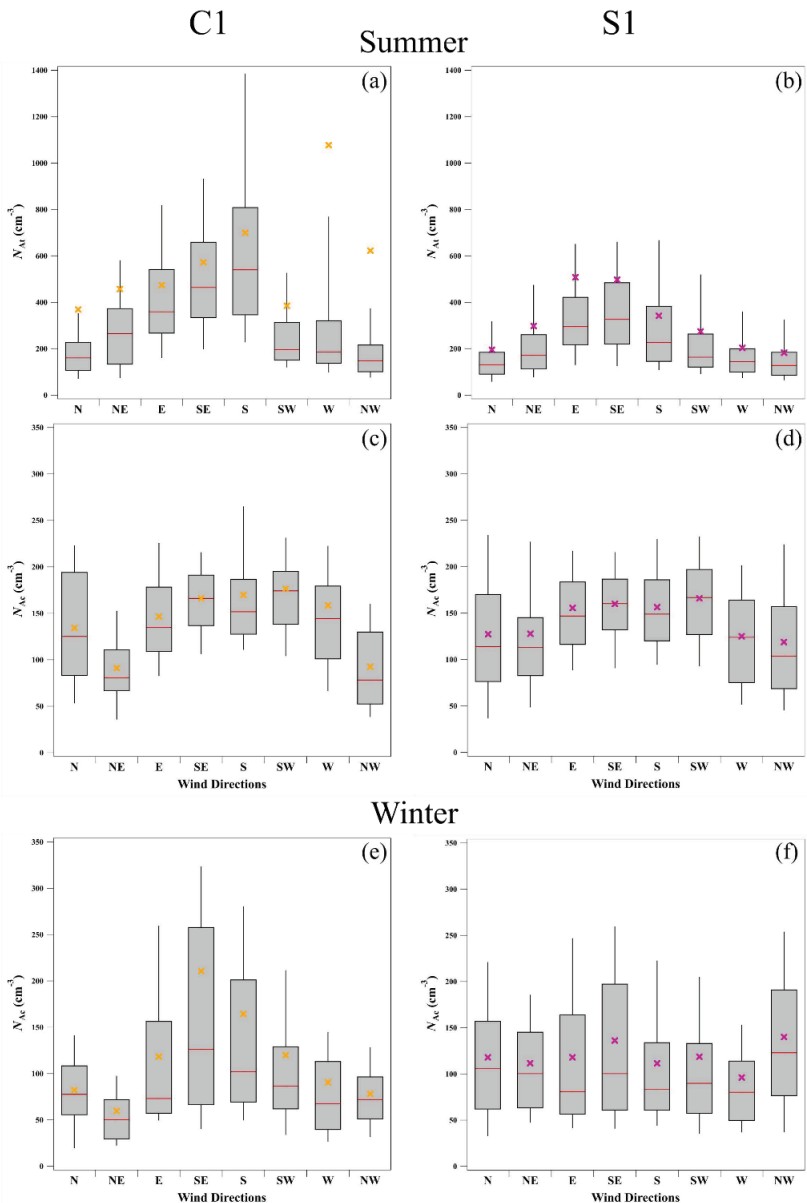

**Figure 6.** $N_{At}$ (a,b) in the summer and $N_{Ac}$ in the summer (c, d) and in the winter (e, f). Data is plotted as the function of wind direction (median red lines, mean x coloured by site) at C1 (orange) and S1 (pink). Box bottom: 25%, box top: 75%, whisker bottom: 10%, whisker top: 90%).

$N_{At}$ at C1 (Fig. 6a) had the highest deviation between the mean and median when the wind was coming from the west (mean: 1007 particles cm$^{-3}$, median: 185 particles cm$^{-3}$) and northwest (mean: 623 particles cm$^{-3}$, median: 148 particles cm$^{-3}$). $N_{At}$ at C1 from the west and northwest are associated with the direction of the airport runway. Aircraft produce submicron aerosol of different mean $D_p$ during different modes of operation. Distinct aerosol size distributions centered at ~ 90 nm from nearby aircraft during landing and take–off, while a sub–30 nm mode has been observed to be prevalent during periods when aircraft are idling on the ground (Herndon et al., 2005). While we were not able to resolve these differences in the ENA dataset due to the lack of size information below 70 nm, we were able to confirm that the largest variability was observed in the smallest



mode of particles shown here, $N_{At}$, when the wind was from the directions associated with the airport and its operation. $N_{At}$ from the direction of the road to the airport, east to south, at C1 was not observed to have significantly higher variability that the other directions.

$N_{At}$ at S1 had less variability between the mean and median when averaged over all wind directions in comparison to C1 in the summer (Fig. 5a). The highest variability in the data at S1 was associated with wind directions from the east (mean: 507 cm$^{-3}$, median: 294 cm$^{-3}$) and southeast (mean: 498 cm$^{-3}$, median: 326 cm$^{-3}$). The road and pasture were to the east and the airport runway, terminal and parking lot were to the southeast of S1 (see SI for more information).

Winter $N_{At}$ at C1, not shown, exhibited a similar trend in regard to the summer data with the highest $N_{At}$ coming from the west. $N_{At}$ from the west had the highest mean of all the wind directions at 1650 cm$^{-3}$ with a corresponding median an order of magnitude lower at 170 cm$^{-3}$. The deviation between the median and the mean was the greatest from the west during the winter, approximately a factor of two times greater than what was observed in the summer. The deviation in the mean and median when the wind was coming from the northwest was smaller than what was observed in the summer at C1. $N_{At}$ was not available
at S1 in the winter for comparison with C1, although we expect it would have had less variability.

As was observed in Fig. 5, $N_{Ac}$ represented a smaller fraction of the total submicron aerosol at C1 and S1 with a lower variability between the median and mean $N_{Ac}$. Mean and median $N_{Ac}$, in Fig. 6c, d, had a low variability across all wind directions at C1 and S1 in the summer. This is in contrast to what was observed for $N_{At}$ during the same period. Mean $N_{Ac}$ were between 92 and
170 cm$^{-3}$ at C1, and the median $N_{Ac}$ were between 78 and 174 cm$^{-3}$. Mean $N_{Ac}$ were between 118 cm$^{-3}$ and 165 cm$^{-3}$ at S1 in the summer (Fig. 6d). The median number concentrations are between 103 cm$^{-3}$ and 166 cm$^{-3}$.

The largest variability of $N_{Ac}$ at C1 was observed in the winter when the wind was from the southeast (mean: 210 cm$^{-3}$, median: 126 cm$^{-3}$) and south (mean: 164 cm$^{-3}$, median: 102 cm$^{-3}$) as is shown in Fig. 6e. This variability was significantly below what
was observed for $N_{At}$ as a function of wind direction in the winter. Winter $N_{Ac}$ at S1 (Fig. 6f) had more variability between the median and mean $N_{Ac}$ than what was observed in the summer. It was still significantly less than what was observed in the $N_{At}$ in the summer. The largest variability in $N_{Ac}$ at S1 in the winter was observed when the wind was from the east (mean: 123 cm$^{-3}$, median: 80 cm$^{-3}$) and southeast (mean: 140 cm$^{-3}$, median: 96 cm$^{-3}$). S1 data when compared to C1 had less variability across all wind directions and seasons, similar to what was observed when comparing $N_{At}$ at C1 and S1.

In summary for the data shown here, $N_{At}$ exhibited the highest variability represented as a high bias of the mean versus the median of all the submicron modes. The highest bias in the mean values in comparison to the medians was associated with the direction of airport operations at C1 (north and northwest) and S1 (east, southeast and south). At ENA the high $N_{At}$ variability was most likely due to local combustion sources based on the size and the wind directions from which they were observed.
This conclusion is supported by the fact that combustion sources are known to produce high concentrations of small mode particles with $D_p < 200$ nm. The high variability observed at ENA was mostly confined to the $N_{At}$, although was also observed in $N_{Ac}$ during the winter. The main regional sources of $N_{Ac}$ at ENA have been attributed to the entrainment of aerosol from the free troposphere and the growth of $N_{At}$ (Zheng et al. 2018). As such, the variability observed here within $N_{Ac}$ in the winter likely includes these processes and local aerosol sources that were not observed in the summer. Chemical and optical property
measurements collected by the AOS should be used in the future to further validate the aerosol sources associated with the variability observed here in the summer and winter.





### 4.4 High time-resolution data

Time series of $N_{tot}$ at C1 and S1 indicated that both locations periodically sample high concentrations over time periods < 4 minutes. High $N_{tot}$ such as these are typically the result of local sources due to their high concentrations and short durations which would become less evident at greater distances from the source. Since aircraft idling, taxiing, take-off, and landing are

all potential times when high $N_{tot}$ could be sampled at C1 and S1, we use the Graciosa airport flight logs and the AOS camera observations to validate high time-resolution $N_{tot}$ data at ENA.

In Fig. 7, we present two four-day periods sampled at C1 and S1 during the summer. $N_{tot} > 25,000$ particles cm$^{-3}$ were observed on a daily basis at C1 in the raw one-second data (Fig. 7a). Lower $N_{tot}$ daily maximum concentrations > 11,000 cm$^{-3}$ were

observed at S1. Winter $N_{tot}$ daily maximums at C1 were > 20,000 cm$^{-3}$ with maximum concentrations occasionally ~80,000 cm$^{-3}$. Figure 8b shows a time period when the overall trend is the reverse of Fig. 7a when higher $N_{tot}$ were observed at S1 in comparison to C1. While this period did not represent the overall trend in $N_{tot}$ between C1 and S1, it was included to show that C1 and S1 both observed $N_{tot}$ maximums at different times and that both sites are impacted by high concentration aerosol events in high time-resolution.

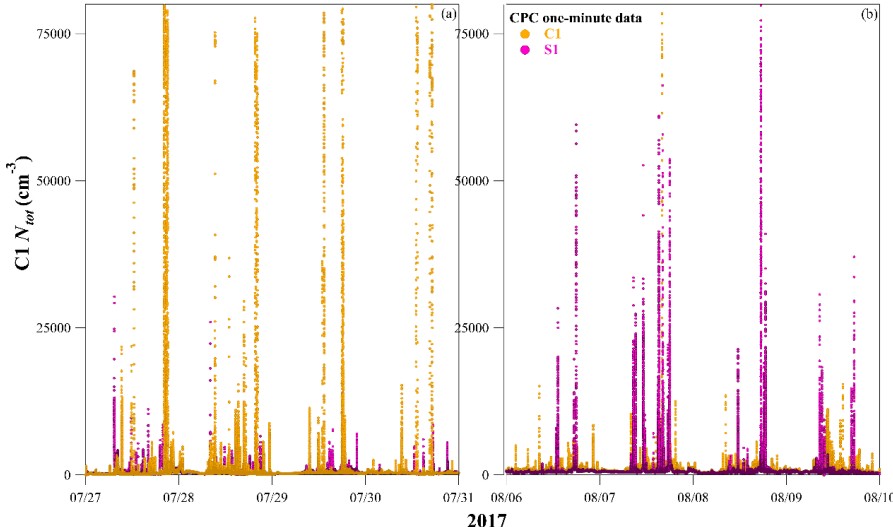

**Figure 7**. Two four-day time periods at C1 (orange) and S1 (pink) raw one-second $N_{tot}$ in the summer.

Graciosa airport on average hosts two flights a day, the first typically in the late morning/early afternoon, and the second in

the late afternoon. The airport time tables for 2017 and 2018 reported that planes landed and took off from Graciosa Island during three distinct time periods throughout the day: ~17% of the planes arrive at Graciosa airport between 8:30 and 11:00 UTC, ~26% between 13:00 and 15:00 UTC, ~56% between 17:00 and 20:00 UTC. Taking into account the wind direction, planes typically land from the east and take off from the west. We confirmed that, during the summer, 97% of the flights occurred in this direction by analysing the daily video from the AOS cameras at C1. However, due to the runway's limited

length, planes often utilize the full length of the runway, which was observed in $N_{tot}$ at C1 and S1. Such occurrences affected $N_{tot}$ at C1 the most when the wind direction was between northeast and west, and S1 when the wind came from the east to southwest.





To further understand the potential influence of the airport operations on $N_{tot}$ at C1 and S1, we examined a one-day time period in detail. In Fig. 8, we present C1 and S1 one-minute time-resolution $N_{tot}$ on August 3, 2017. $N_{tot}$ at S1 was largely unaffected by the short duration high concentration aerosol events as $N_{tot}$ never was > 1000 cm$^{-3}$. While this is only a one-day time period and was by no means representative of daily $N_{tot}$, we show it as an example of the complexity within $N_{tot}$ at ENA.

Throughout the day, abrupt changes in wind direction were observed. Winds from the south, southwest and west dominated until 17:58 UTC. Starting at 18:00 UTC, the dominant wind directions were northwest, north and east. Analysis of the video from the AOS camera at C1 showed that diesel trucks were on the runway from 09:07 UTC to 09:27 UTC for daily maintenance. At two times during the afternoon, 13:42 to 15:02 UTC and 18:46 to 19:51 UTC, the aircraft was idling near the

airport terminal (Fig. 8). During the first part of the day, when the wind directions were from the south and west, $N_{tot}$ > 1000 cm$^{-3}$ at C1 at numerous times. In the SI we identify these directions at C1 with the airport terminal, parking lot, and the road to the airport. Later in the day, when the winds are coming from the northwest to east, in the direction of the runway at C1, $N_{tot}$ < 1000 cm$^{-3}$ at C1 similar to $N_{tot}$ at S1.

The high $N_{tot}$ events at C1 and S1 were associated with the airport activities and increased road traffic that generally occurred before the arrival and after the departure of the aircraft, based here on visual observations, airport flight logs, and wind degree direction analysis. The aircraft and vehicle impacts were observed by sharp peaks occurring on timescales on the order of minutes when $N_{tot}$ was an order of magnitude above the baseline signal. In contrast, the airport operations tended to cause periods of elevated $N_{tot}$ that occurred over longer timescales on the order of hours. Therefore, the impact of the airport, its

operation and associated traffic on the AOS data at ENA could not be constrained to the arrival and departure times of the aircraft since it was also impacted by airport operations that occurred throughout the day and the wind direction in relation to C1 and S1.

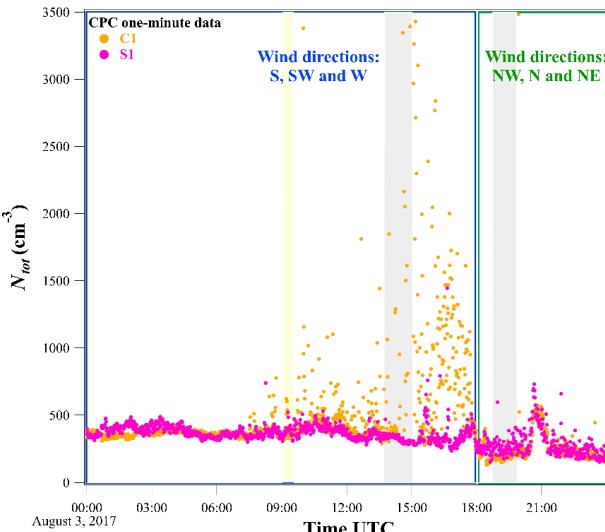

**Figure 8**. $N_{tot}$ at C1 (orange) and S1 (pink) on August 3, 2017. Yellow and grey periods indicate when the AOS cameras observed trucks on the runway (yellow) and planes near the terminal building (grey). Blue and green boxes indicate the dominant wind directions at the time.



While the influence of the airport operations may not be readily apparent from the short duration high concentrations observed at C1 and S1 (Fig. 7), further information constraining this influence is obtained by looking at the diurnal cycle of mean and median $N_{tot}$ at C1 (Fig. 9).

The three hourly periods with highest mean $N_{tot}$ were observed during 9:00 to 10:00 UTC at 916 cm$^{-3}$, 13:00 to 14:00 UTC at 860 cm-3, and 17:00 to 18:00 UTC at 1,595 cm$^{-3}$. These three elevated mean $N_{tot}$ periods occur during the three times when the airport flight logs on average was observed to host flights. These periods were identified using the airport flight logs and are shown as the black boxes in Fig. 9. Mean $N_{tot}$ from 7:00 to 8:00 UTC reached a value of 615 cm$^{-3}$. The highest mean $N_{tot}$ was observed during the third time period identified by the airport to host on average half of the daily flights, while the two
earlier time periods were only associated with ~25% of the flights each. Two other high mean $N_{tot}$ periods were observed during the diurnal profile at C1. Mean $N_{tot}$ were 615 cm$^{-3}$ from 7:00 to 8:00 UTC, that occurred during a similar time that the AOS cameras observed the daily maintenance of the runway with diesel trucks from 7:45 and 8:30 UTC. The second period from 20:00 to 21:00 with mean $N_{tot}$ > 800 cm$^{-3}$ was attributed to known potential aerosol sources at this time.

The diurnal variation observed in the mean $N_{tot}$ at C1 in the summer was not observed in the median $N_{tot}$. Hourly averaged medians exhibited low variability throughout the day with a minimum of 380 cm$^{-3}$ during the night between 23:00 and 24:00 UTC. A maximum of 506 cm$^{-3}$ was observed in the late afternoon between 17:00 and 18:00 UTC.

At ENA, the periods with the largest deviation between the median and mean $N_{tot}$ were the three periods when most of the
flights occur at the airport. A diurnal variation was observed in the mean $N_{tot}$, yet was not statistically relevant for the median $N_{tot}$ of the same data at C1 and S1. While not shown here, S1 had a similar trend in the diurnal profile to what was observed at C1 in the summer. The main difference was that the mean $N_{tot}$ were all < 1,000 cm$^{-3}$. Winter data at C1 also had the highest mean $N_{tot}$ and their deviations from the medians during the hours of airport operations. We use the information from the diurnal profiles at ENA to validate the statement that the airport operations and associated activities were the largest sources of high
concentration $N_{tot}$ observed at ENA.

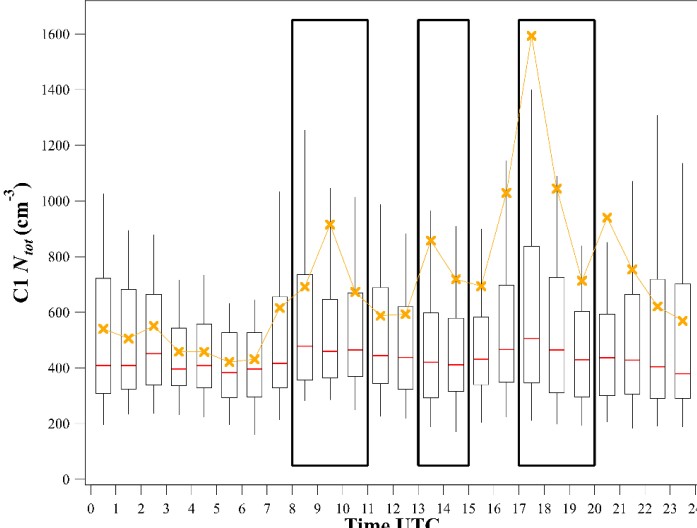

**Figure 9**. Box and whisker diurnal profile of N$_{tot}$ at C1 during summer. N$_{tot}$ mean (orange x) and median (red line). Black boxes from 08:00-10:00, 13:00-15:00 and 17:00-20:00 UTC indicate the three daily time periods when aircraft were present at the Graciosa airport.



### 4.5 High number concentration aerosol event mask

#### 4.5.1 Algorithm parametrization

To apply a mathematical algorithm to mask high $N_{tot}$ events at C1 and S1, we first calculated the $\sigma_b$. We found $\sigma_b$ values of

298 cm⁻³ and 264 cm⁻³ for C1, respectively in the summer and winter, and $\sigma_b$ values of 234 cm⁻³ for S1 in the summer. Then, we conducted a sensitivity test to select the optimal parametrization of the algorithm to apply to the one-minute resolution $N_{tot}$ data at ENA using the combination of the α parameter and thresholding method shown in Table 1, Section 3.2. We use one-month time periods as the utilization of longer periods can bias the characterization of the regional baseline due to seasonality, which can accentuate the long-term variabilities and confuse the high signal of local events (El Yazidi et al., 2018). Previous

studies use the random walk (RW) threshold for aerosol data. Drewnick et al. (2012) proposed using α = 3 to remove sharp and short peaks lasting a few seconds in $N_{tot}$ from the CPC and gas-phase CO measurements from a mobile aerosol research laboratory. The authors found that the application of the α3-RW parameterization worked well when the density of the high concentration events was low. Similarly, El Yazidi et al. (2018) used α1-RW with gas-phase $CO_2$ and $CH_4$ data at four different stations in Europe that were affected by sharp events over time periods of a few minutes. The α1-RW parameterization was

able to detect ~96% of the events that were visually identified by the station manager for their data. Therefore, we began at ENA by testing two α values with the RW threshold that were used in these two studies.

We present, in Fig. 10, the results from the application of the algorithm over the same twenty-four hour period that we analyzed in Section 4.4, Fig. 8. The first two parameterizations selected, α1-RW and α3-RW, were able to identify the first data points

during a high $N_{tot}$ event, but were not able to identify events that occurred for extended periods of time on the order of hours, as is shown in Fig. 10a for the application of α3-RW at C1. While α1-RW is not included in the figure for simplicity, similar results were produced from this parameterization. Next, we constrained the threshold more by applying the TP method with the same α values. For both C1 and S1 sites and seasons, the α1-TP parameterization was the only parameterization able to identify longer duration events that lasted from minutes to hours, as were experienced due to airport operations as shown in

Fig. 10b. Results from α3-TP were not included in the figure as the combination of the relaxed α and constrained two point threshold parameters, α3-TP, yielded similar results to the RW threshold parameterizations tested previously. The α3-TP parameterization was not able to identify the longer duration high $N_{tot}$ events. In conclusion, when high $N_{tot}$ events had durations on the order of hours, the difference in the signal between the adjacent points was not high enough to be identified by either the RW threshold or the higher α=3 parameter combinations tested at C1.



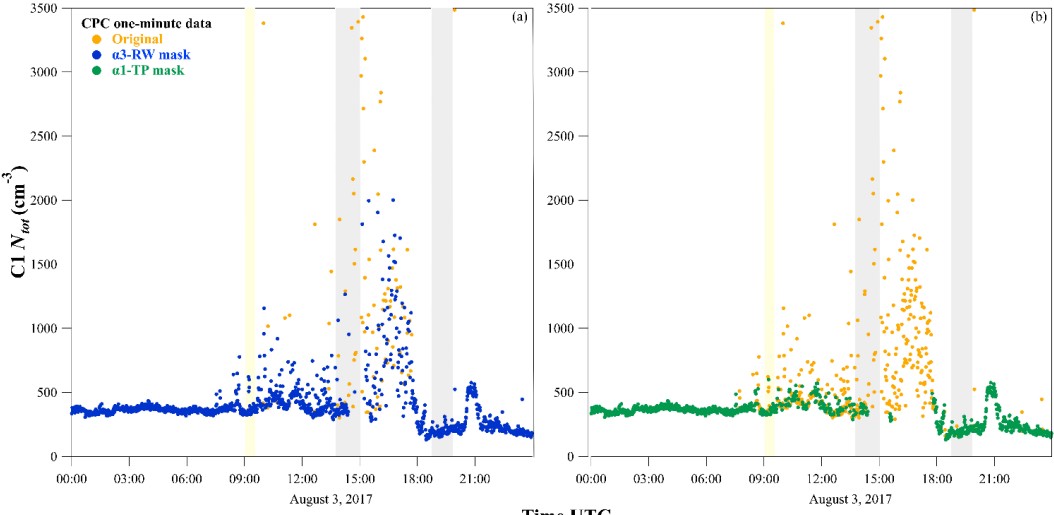

**Figure 10.** Original (orange points) and masked $N_{tot}$ at C1 using (a) α3-RW (blue points) and (b) α1-TP (green points) parameterizations over a twenty-four hour period on 8/3/2017. Yellow and grey boxes indicate periods when the AOS cameras at C1 detected trucks and planes, respectively, on the runway.

Similar results were obtained when we tested the four parameterizations of the algorithm on $N_{tot}$ at S1, not shown. As we observed at C1, the tightest combination of parameters, α1-TP, was able to most accurately identify all of the high $N_{tot}$ events of all the parameterizations tested here. The higher α values and the random walk threshold relaxed the algorithm such that the number of data points identified was likely to underestimate the number and duration of high $N_{tot}$ events observed at ENA.

Therefore, we used the α1-TP parameterization to create an aerosol mask at ENA, heretofore referred to as as the ENA Aerosol Mask (ENA-AM), using $N_{tot}$. At ENA the one-minute $N_{tot}$ had sufficient time resolution to mask the high $N_{tot}$ events. Application to the higher time-resolution one-second $N_{tot}$ data was not necessary based on the validation of ENA-AM presented here and saves computational time when analyzing continuous data.

Furthermore, we evaluated the ability of ENA-AM to mask short-lived high $N_{tot}$ events during periods when ENA was sampling long-range transported aerosol. Periods with elevated aerosol concentrations due to long range-transported continental sources occur at ENA for durations on the order of days to weeks. Through the analysis of back trajectories and aerosol optical properties, we identify and present an episode of transported aerosol from Central Africa and the Canary Islands from January 7 to 12, 2017 (Fig. 11a). For several days during this time, $N_{tot}$ at ENA remained above 700 cm⁻³, likely due to a mixture of

mineral dust and black carbon from biomass burning sources (Logan et al., 2014), as have been observed from other continental sources at ENA (Zheng et al., 2018). After applying ENA-AM to $N_{tot}$ at C1, we observed that the majority of the data associated with the multiday event were retained with the baseline $N_{tot}$. Simultaneously, the short duration high $N_{tot}$ events, attributed to local sources, were removed (Fig. 11b). We use the results from this case study to validate the application of ENA-AM to one-minute $N_{tot}$ during periods when multi-day entrained long-range transported aerosol were sampled at ENA.



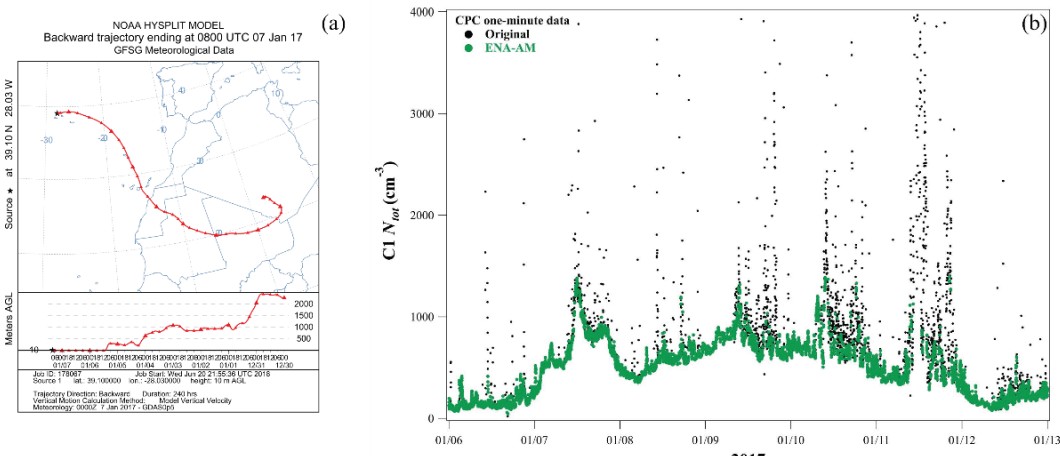

**Figure 11**. An episode of long-range transport of continental air masses at C1 determined (a) with an 8-day back trajectory arriving at 10 m a.g.l. and (b) $N_{tot}$ during the event with original (black) and masked (green) data using ENA-AM.

Since C1 may experience higher concentrations from local sources that have become more dilute within the region due to proximity than S1 (see SI S1), we reduced α to a value of 0.5 in attempt to remove such influences. This approach increased the amount of data that was masked, removing 47% (C1) and 34% (S1) versus 26% (C1) and 15% (S1) with ENA-AM. A linear regression R-squared ($R^2$) between α = 0.5 masked C1 and S1 also yielded a lower slope (slope = 0.76 ± 0.002, $R^2$ = 0.90) indicating a larger discrepancy between the two sites. The algorithm was also no longer able to discriminate between variations in the baseline due to regional process (e.g. entrainment of particles from the free troposphere due to long-range transport events shown in Fig. 11) from weak local aerosol events related to unattributed local sources. Therefore, reducing α at C1 was not effective for removing longer duration smaller $N_{tot}$ variability at ENA.

### 4.5.2 Identification of a regional baseline and impact of ENA-AM on $N_{tot}$ and $N_{Ac}$

With the application of ENA-AM at two locations, we were able to mask high $N_{tot}$ events and to define a regional baseline for $N_{tot}$ at ENA. In Fig. 12, we show original and masked $N_{tot}$ at C1 and S1 during the summer. Due to the diverse high $N_{tot}$ events and local sources at ENA, the $R^2$ value of the original $N_{tot}$ between the two sites (black dots) was minimal ($R^2$ = 0.03). After applying ENA-AM to both datasets (green dots), 26% data were masked at C1 and 15% at S1. Similarly, 15% of the data at C1 in the winter was masked. Furthermore, the linear regression generated an $R^2$ = 0.87 with a slope = 0.84 ± 0.001 (data was fit through zero). C1 retains a higher $N_{tot}$, likely due to the incomplete removal of sources discussed in 4.5.1 and further in SI. The variability in the original $N_{tot}$, due to high concentration aerosol events was removed and a regional baseline was identified based on the agreement between the two locations within measurement and mask uncertainties.





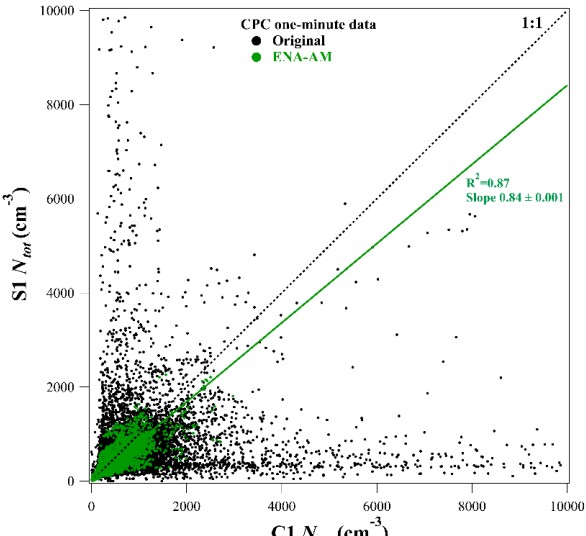

**Figure 12**. Scatter plot of $N_{tot}$ at C1 and S1 in the summer. Original one-minute data (black) is shown with masked data employing ENA-AM (green).

After the application of ENA-AM, we observed that mean, deviation between mean and median, and standard deviation $N_{tot}$ all experienced reductions. In Table 3 we show mean, median and standard deviations for the original and ENA-AM masked $N_{tot}$ and $N_{Ac}$ measurements at C1 and S1 in the summer and winter. Applying ENA-AM to C1 $N_{tot}$, mean and standard deviation values dropped from $707 \pm 2780$ cm$^{-3}$ to $428 \pm 228$ cm$^{-3}$ in the summer and from $537 \pm 630$ cm$^{-3}$ to $347 \pm 223$ cm$^{-3}$ in the winter. At S1, the decrease was lower yet still significant, from $489 \pm 370$ cm$^{-3}$ to $384 \pm 355$ cm$^{-3}$. In the summer, ENA-AM

mean $N_{tot}$ was 9.1% higher at C1 than at S1. Satellite images and analysis of local aerosol sources (see SI) show that C1 is located $\sim$ 1 km closer to urbanized areas and to the town of Santa Cruz than S1. The $N_{tot}$ generated from these more distant and diffuse sources is likely too weak to be completely masked by ENA-AM as discussed in Section 4.5.1.

Contrarily to $N_{tot}$, in the summer, $N_{Ac}$ mean, median and the deviation between them remained largely unchanged after the

application of ENA-AM. This is in agreement with Section 4.3.2, where we showed summer $N_{Ac}$ to be only minimally affected by local aerosol events. However, in the winter, mean, deviation between mean and median, and standard deviation $N_{Ac}$ at C1, experienced a higher reduction when masked with ENA-AM (25% for the mean, 51% for the deviation between mean and median, and 73% for the standard deviation). These results are likely related to the presence of additional sources in the winter (e.g. burning of wood for home heating) which might affect $N_{Ac}$ in a way that is not masked by ENA-AM.

**Table 3.** Mean, median and standard deviations of original and ENA-AM masked one-minute $N_{tot}$ at C1 and S1 during the summer and winter.

| | | Summer C1 | | | Summer S1 | | | Winter C1 | | |
|---|---|---|---|---|---|---|---|---|---|---|
| | | Original | ENA-AM | Reduction | Original | ENA-AM | Reduction | Original | ENA-AM | Reduction |
| $N_{tot}$ (cm$^{-3}$) | **Mean** | 707 | 428 | 39% | 489 | 384 | 21% | 537 | 346 | 36% |
| | **Median** | 427 | 387 | 9% | 370 | 355 | 4% | 366 | 290 | 21% |
| | **St. Dev.** | $\pm 2780$ | $\pm 228$ | 92% | $\pm 1012$ | $\pm 193$ | 81% | $\pm 630$ | $\pm 223$ | 65% |
| $N_{Ac}$ | **Mean** | 160 | 150 | 6% | 151 | 149 | 1% | 140 | 105 | 25% |
| | **Median** | 156 | 152 | 3% | 147 | 147 | - | 91 | 81 | 11% |





| St. Dev. | ± 142 | ± 60 | 58% | ± 79 | ± 75 | 5% | ± 296 | ± 79 | 73% |
| --- | --- | --- | --- | --- | --- | --- | --- | --- | --- |

To estimate the influence of local aerosol events on daily $N_{tot}$ and $N_{Ac}$, we investigated the deviation between the original and ENA-AM masked $N_{tot}$ and $N_{Ac}$ daily means at C1 in the summer and winter in Figure 13. We observed that after applying ENA-AM, depending on the day, $N_{tot}$ daily means experienced reductions varying between 7% and 81% in the summer and between 2% and 67% in the winter. $N_{Ac}$ reductions were lower than 27% in the summer and 40% in the winter (with the exception of two days, December 16 and 22, when $N_{Ac}$ daily means experienced reductions of 61% and 80%). As already observed in Section 4.3.2, at ENA, especially in the summer, local sources are mainly responsible for the emission of small particles in the At mode, while the Ac mode is generally only minimal impacted. Thus, in general, the reduction in $N_{Ac}$ after masking the data does not impact the daily mean values. The higher daily mean reductions observed for $N_{tot}$ in comparison to $N_{Ac}$ after the utilization of ENA-AM, demonstrated that the algorithm was able to selectively detect and isolate the measurements impacted by local aerosol events without having to use size distribution data. The high original $N_{tot}$ and $N_{Ac}$ daily means and the large deviation after application of ENA-AM (80%) observed on December 22 were exceptions likely related to a poor-efficiency combustion source, a bulldozer, not normally present at C1 that was observed by the AOS cameras. The time series plots highlight $N_{Ac}$ up to 11,000 cm$^{-3}$ between 14:50 and 18:30 UTC. Thus, while $N_{tot}$ and $N_{Ac}$ measurements were impacted, ENA-AM was able to mask the data.

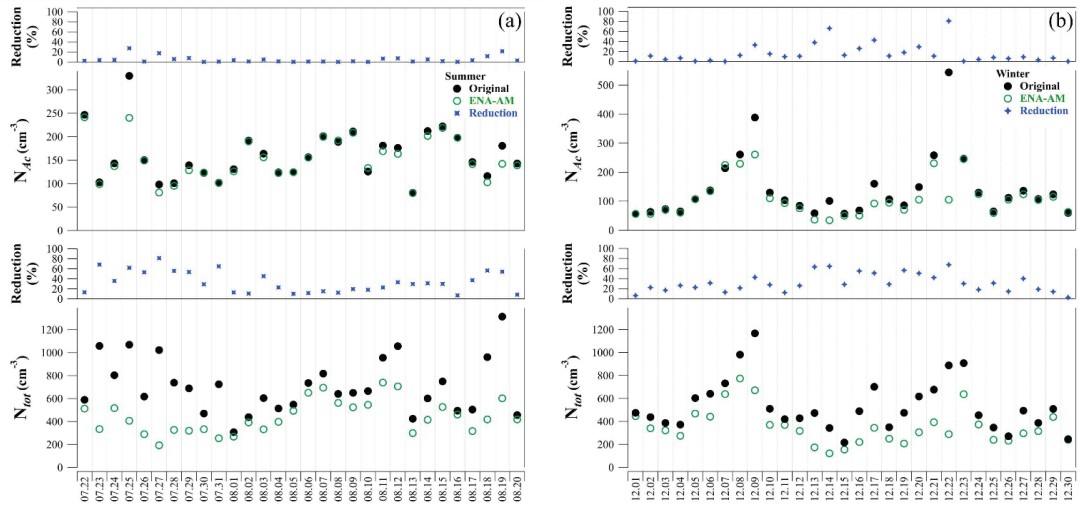

**Figure 13.** Original (black) and ENA-AM masked (green) $N_{tot}$ and $N_{Ac}$ daily means at C1 and corresponding reduction (%) (blue) in the (a) summer and (b) winter.

### 4.5.3 Comparison of ENA-AM to other masks

We tested using wind direction to mask local aerosols by applying a meteorological mask to remove C1 and S1 $N_{tot}$ measurements as a function of the wind directions associated with the airport (west to northwest and southeast to south at C1 and east to south at S1). After applying the meteorological mask at C1, 38.9% in the summer and 62% in the winter of the AOS data was removed. Similarly, at S1 43.4 % of the data in the summer were masked. Only 9.8% of the C1 and S1 $N_{tot}$ datasets remained for comparison between C1 and S1, which limited our ability to to determine the regional background. The linear regression generated an R$^2$ of 0.18, likely due the paucity of data. Therefore, masking AOS data based on wind direction resulted in the rejection of too much data to define a regional baseline aerosol.





Masking AOS data at ENA utilizing the associated metadata, such as AOS motion-activated cameras and airport flight logs, was not able to identify all of the periods impacted by local aerosol sources. However, analysis of videos and airport flight logs were useful to confirm the presence of the aircraft at the airport to validate the application of an aerosol mask. These observations and metadata were therefore used to validate the application of ENA-AM as discussed in Section 3.2.

We tested a different mathematical algorithm to filter aerosol data based on previous work by Hagler et al. (2012). The authors applied the coefficient of variation algorithm to ultra-fine particle concentrations and greenhouse gas data. At ENA, this method masked the dominant fraction of the data, 72% at C1 in the summer. We were not able to validate the additional reductions in comparison to ENA-AM with other observations or collocated measurements. Periods with known long-range

transported aerosol were also removed. Therefore, the application of this method was not pursued further at ENA.

Comparison was also made between ENA-AM and the one-second time base filter developed by Zheng et al. (2018) at ENA. Conducted at C1 over two three-month periods in the summer (June to August 2017) and winter (December 2016 to February 2017), the authors found similar baseline values for $N_{tot}$ measurements ($513 \pm 314$ cm$^{-3}$ in the summer and $383 \pm 300$ cm$^{-3}$ in

the winter). They also reported similar $N_{Ac}$ mean and standard deviation values ($143 \pm 81$ cm$^{-3}$ in the summer and $92 \pm 89$ cm$^{-3}$ in the winter) after the additional step of lognormal fitting the size distributions to what we report here with ENA-AM. We validate the original Zheng et al. (2018) algorithm with the additional Supplementary site data using the data from our summer ENA-AM period. We recreated the Zheng et al. (2018) mask on the original one-second time base and applied it to our summer period at C1. We found that it agreed with ENA-AM 68% of the time (see section S2 in the SI). ENA-AM removed less data

than the Zheng et al. (2018) method when mapped onto a one-minute time base (26% versus 41%). ENA-AM was also developed to operate on a longer time base (one-minute versus one-second) to reduce computational requirements.

### 4.5.4 Masked AOS data and AAF Overflights

After determining the regional baseline for $N_{tot}$ from the ground AOS measurements at ENA, we compare ENA-AM masked

C1 $N_{tot}$ with AAF $N_{tot}$ data in Figure 14. We restricted our comparison of $N_{tot}$ from the AAF to an area within a 10 km diameter box centered at C1 at altitudes $\leq 500$ m. Before applying ENA-AM, the R$^2$ value obtained from comparing the original C1 $N_{tot}$ and AAF measurements from seven overflights (data not shown) was poor (R$^2 = 0.26$). After applying ENA-AM at C1, we obtained an R$^2$ of 0.71 and a slope of $1.04 \pm 0.01$, which indicated a good agreement between the AOS and AAF data. The largest deviations from the 1:1 line occurred during two flights, on June 21 (RF1) and June 29 (RF6). On June 21 the AAF

flew over Graciosa Island at two distinct times (12:00 and 13:30) during the day that we represent as two different periods in Fig. 14. While the AAF and C1 $N_{tot}$ data fell on the 1:1 line within measurement uncertainties for the first period during RF1, C1 sampled an average of 43% more $N_{tot}$ than the AAF during the second flyover. The second flyover might have coincided with a period of time when C1 was affected by local events that ENA-AM was not able to identify and mask, as discussed in the paragraph above. Unfortunately, due to a lack of data at S1 during this time, this could not be verified. The largest deviation

from the 1:1 line of all flights was observed on June 29 when higher $N_{tot}$ was observed by the AAF. The mean $N_{tot}$ was $659 \pm 17$ cm$^{-3}$ at C1 and $1,141 \pm 828$ cm$^{-3}$ at AAF. The flight trajectory indicated that the AAF flew south of C1 over the center of the island and around the town of Santa Cruz and two smaller towns, Guadalupe and Vitoria, on an ascending path. The AAF $N_{tot}$ measurements might be affected by local aerosol when the AAF flew over these urbanized areas. AAF $N_{tot}$ measurements might also have been biased by AAF emissions sampled through the aerosol inlet while the aircraft was gaining altitude. The

standard deviation of the AAF data was also significantly greater than what was observed at C1, indicating the AAF likely intercepted plumes not observed at C1 during this time. Further analysis of the AAF data from RF6 would be required to determine the source of the discrepancy with the AOS data.



The high R$^2$ obtained from the masked C1 and AAF $N_{tot}$ demonstrated that aerosol in the summer were well-mixed within the first 500 m of the marine boundary layer. This is likely due to the high sea level pressure system and advection in the summer at ENA which might enhance submicron aerosol mixing within the MBL (Davis et al., 1997). Since our focus here was on

summer data at ENA due to the deployment of S1 to constrain C1 measurements, the winter season and vertical characterization of the MBL was beyond the scope here. During the winter, aerosol in the MBL are expected to be less well-mixed due to a strong polar front activity and low pressure system (Barbosa et al., 2017). Therefore, we expect less correlation between the AOS data and the AAF in the winter than what was observed in the summer.

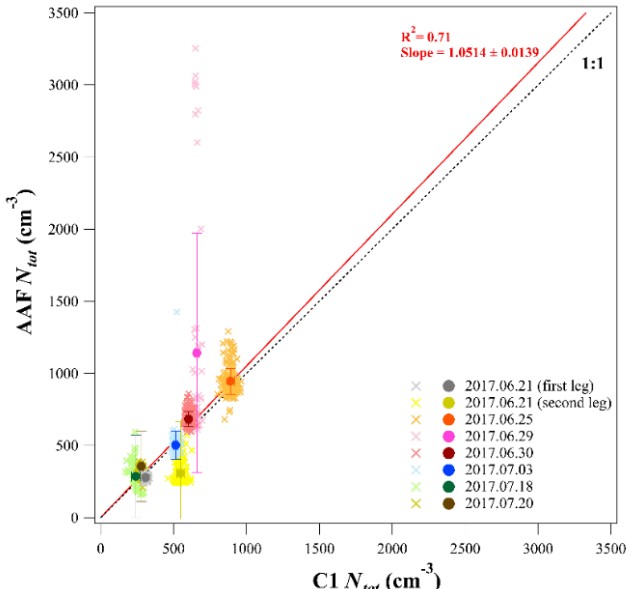

**Figure 14.** Scatter plot of ENA-AM masked C1 $N_{tot}$ and AAF $N_{tot}$ collected within 10 km from C1 and at altitudes ≤ 500 m.

## 5 Conclusions

High concentration aerosol events were observed in the AOS data at the ENA Central Facility. Analysis of the submicron aerosol concentrations and size distributions were used with collocated meteorological data (wind direction) to associate high

concentration aerosol events with potential local aerosol sources. Total submicron and Aitken mode aerosol were the most affected as determined by wind direction and should be masked before conducting ambient aerosol process studies at ENA. Accumulation mode aerosol was less impacted, especially in the summer. Ac mode might then be used without applying an aerosol mask as representative of the regional aerosol.

We developed a novel aerosol mask at ENA called ENA-AM and validated its application by using two measurement locations located within 1 km of each other. The temporary Supplementary site was deployed to validate the new aerosol mask at the Central Facility with the AOS. Time periods impacted by high concentration aerosol events were removed, and we were able to define a regional baseline for the submicron number concentration data at ENA during the summer and winter. The masked submicron aerosol number concentrations from the ground site were compared with the AAF aircraft data during flights over

the facility. It was possible to determine a well-mixed regional aerosol within the first 500 m of the marine boundary layer for the data presented here collected during the summer ACE-ENA IOP.



Application of ENA-AM required measurements in which: 1) the time resolution of the dataset was shorter than the typical length of the event, and 2) the variation within the baseline data was smaller than the variation during the periods containing local aerosol. The CPC one-minute submicron number concentration data satisfied these requirements at ENA. Therefore, we

developed an algorithm using the CPC data at ENA that could be applied to the AOS data for studying regional aerosol processes. After the application of ENA-AM, 26% of the one-minute data at C1 and 15% at S1 were masked in the summer. ENA-AM masked a lower percentage of the data than the wind direction mask, which masked 38.9% of the data at C1 data and 43% at S1. Compared to the meteorological method, ENA-AM removed approximately half of the data than the mask based on wind direction and, more importantly, resulted in a higher $R^2$ between the sites, 0.87 versus 0.18. Minimal deviations

between the original median $N_{tot}$ and ENA-AM mean values at C1, respectively, were 427 and 428 particles cm$^{-3}$ (summer), and 370 and 384 particles cm$^{-3}$ (winter). Therefore, it is possible that median values might be used to study longer term trends in the data without applying an aerosol mask. While useful, for example, to study seasonal trends, this approach would not be suitable for studying short time period aerosol variability on the order of minutes to hours as is required in ambient aerosol process research. For this reason, application of an aerosol mask such as ENA-AM is recommended even at remote locations,

when studying high time-resolution submicron aerosol processes, especially those focused on Aitken mode particles. Application of ENA-AM, or other aerosol masks, is possible at other locations with AOS or similar data. Validation should include comparison with other collocated measurements, observations and metadata when available.

Data availability: Data used in this study are publicly accessible at the permanent archive of data collected at the Eastern North Atlantic site of the Atmospheric Radiation Measurement (ARM) user facility (available at https://www.archive.arm.gov/discovery).

ACA and FG conceptualized the analysis and wrote the manuscript. FG led the analyses with input from JU, SRS, RW and FM. ACA was the project administrator. All authors were involved in helpful discussion and contributed to the manuscript.

Competing interest: The authors declare that they have no conflict of interest.

Acknowledgements: The work was supported by the Atmospheric Radiation Measurement (ARM) program, funded by the U.S. Department of Energy (DOE), Office of Science, Office of Biological and Environmental Research). We acknowledge the ARM Research Facility, a user facility of the U.S. DOE, Office of Science, sponsored by the Office of Biological and Environmental Research for providing data. We also acknowledge the ENA ARM Site Manager, Heath Powers, Operations Manager, Paul Ortega, and Site Operators, Carlos Sousa, Tercio Silva and Bruno Cunha.




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
