# Peer review of "Identifying a regional aerosol baseline in the Eastern North Atlantic using collocated measurements and a mathematical algorithm to mask high submicron number concentration aerosol events"

_Atmospheric Chemistry and Physics, 2020_

## Referee Comment (RC1) · Anonymous Referee #1 · 28 Feb 2020

An algorithm is developed and presented for removing the influence of local sources from long-term, high time resolution data collected at DOE's User Facility in the Eastern North Atlantic (ENA). The technique should be useful for others interested in performing a similar data analysis. I only have minor comments which are listed below.

Lines 19 – 21: What was the RH of the sample air at C1 and S1? The size ranges of the three optical size modes should be reported at the sampling RH.

Line 18: Change to "….to create AN aerosol mask…"

Figure 2: The axes don't seem to be labeled correctly. The x-axis should be nm, not nm^-3. Is the y-axis dN/dlogD?

Figure 7: It might be easier to see differences between C1 and S1 if one were plotted with slightly larger markers and presented behind the other one.

Line 11: Should be "Figure 8", not 8b.

Figure 8: Wind direction should be plotted on an additional y-axis axis rather than using the vertical bars as an indication of wind direction values.

Line 4: Define $\sigma b$.

Figure 11b: It is hard to differentiate the dark green and black markers. Maybe make the dark green a brighter color?

---

## Referee Comment (RC2) · Anonymous Referee #2 · 13 Mar 2020

This manuscript developed an algorithm (ENA aerosol mask) to identify local emissions in two remote sites with high time resolution particle number concentration data. This algorithm can be useful for other measurements in remote locations but it is not easy for other studies to apply it due to the lack of parameterization tests in the current manuscript. Since the aerosol mask the key point of this manuscript, I recommend the authors to re-organize it so that it can be easier for other researchers to follow and thus achieve a higher impact.

The goal of this study is to filter high-frequency, high-intensity signals statistically. It is only reachable when natural sources are significantly less variable and lower in intensity. Even when that is true, all algorithms will inevitably balance the trade-off between losing real natural variability and including more local anthropogenic influences. The discussion about this trade-off in this manuscript is not very systematic or vigorous. It seems that this method is working well with the dataset but I am not very convinced that the final parameterizations are optimal in this study. See specific comments below:

1. It did not try to compare to other pre-existing methods for this problem, for example, the smoothing methods that can remove the spikes? (Liu et. al. 2018, Velle 1977 and Goring 2002)

Liu, Jun, et al. "High summertime aerosol organic functional group concentrations from marine and seabird sources at Ross Island, Antarctica, during AWARE." Atmospheric Chemistry & Physics 18 (2018): 8571-8587.

Goring, D. G. and Nikora, V. I.: Despiking acoustic Doppler,velocimeter data, J. Hydraul. Eng.-ASCE, 128, 117–126,https://doi.org/10.1061/(asce)0733-9429(2002)128:1(117),2002.

Velleman, P. F.: Robust nonlinear data smoothers – definitions and recommendations, P. Natl. Acad. Sci. USA, 74, 434–436,https://doi.org/10.1073/pnas.74.2.434, 1977

2. Parameterizations of $\sigma b$ and alpha need to be improved. Page 7, Line 6: "We determined the standard deviation of the data below the median ($\sigma b$) of Ntot for each of the two one-month periods." This sounds arbitrary and needs explanation. Is two months the result of a sensitivity check? Is it related to the time resolution of Ntot? I suppose the moving medians are used here for all the data points? Why below-median, not another percentile?

The authors only tested four scenarios in Table 1. Can the authors do a sensitivity test with more data points?

3. The wind direction, wind speed, and size distribution sections are long and not necessarily related to the aerosol mask. Consider shorten them or move part of them to the supplement.

I will support its publication if the authors can address my comments.

---

## Author Comment (AC2) · 14 May 2020

**Manuscript No.**: acp-2020-49

**Title**: Identifying a regional aerosol baseline in the Eastern North Atlantic using collocated measurements and a mathematical algorithm to mask high submicron number concentration aerosol events

5 **Responses to Anonymous Referee #2**

*General Comments:*

*This manuscript developed an algorithm (ENA aerosol mask) to identify local emissions in two remote sites with high time resolution particle number concentration data. This algorithm can be useful for other measurements in remote locations but it is not easy for other studies to apply it due to the lack of parameterization tests in the current*
10 *manuscript. Since the aerosol mask the key point of this manuscript, I recommend the authors to re-organize it so that it can be easier for other researchers to follow and thus achieve a higher impact. The goal of this study is to filter high-frequency, high-intensity signals statistically. It is only reachable when natural sources are significantly less variable and lower in intensity. Even when that is true, all algorithms will inevitably balance the trade-off between losing real natural variability and including more local anthropogenic influences. The discussion about*
15 *this trade-off in this manuscript is not very systematic or vigorous. It seems that this method is working well with the dataset but I am not very convinced that the final parameterizations are optimal in this study.*

**Response:** We thank Anonymous Referee #2 for all of his/her constructive criticism and detailed suggestions. We revised and reorganized the manuscript per the suggestions and agree with Referee #2, including moving some sections, detailed below, to SI. We too agree that now it is overall stronger, more streamlined, and capable of higher
20 impact. We re-organized the discussion section (Section 4.4 of the revised manuscript) to address the concerns about the trade-offs mentioned above when applying the masking algorithm to minimize the loss of natural variability and maximize identification of local anthropogenic sources, which ENA-AM was optimized to do. All detailed comments are addressed in the following point-by-point discussions below. All the alterations to the manuscript are shown in the track changes revised version of the manuscript and Supplemental Information in the
25 attached PDF.

*Detailed Comments:*

**(R2.1)** *It did not try to compare to other pre-existing methods for this problem, for example, the smoothing methods that can remove the spikes? (Liu et. al. 2018, Velle 1977 and Goring 2002)*

30 *Liu, Jun, et al. "High summertime aerosol organic functional group concentrations from marine and seabird sources at Ross Island, Antarctica, during AWARE." Atmospheric Chemistry & Physics 18 (2018): 8571-8587.*

*Goring, D. G. and Nikora, V. I.: Despiking acoustic Doppler, velocimeter data, J. Hydraul. Eng.-ASCE, 128, 117–126, https://doi.org/10.1061/(asce)0733- 9429(2002)128:1(117),2002.*

*Velleman, P. F.: Robust nonlinear data smoothers – definitions and recommendations, P. Natl. Acad. Sci. USA,*
35 *74, 434–436, https://doi.org/10.1073/pnas.74.2.434, 1977*

**[Resp.]:** Thank you for suggesting that we add smoothing methods as an additional method for comparison. We have addressed these concerns thoroughly with discussions in the Introduction and Results Sections of the revised manuscript as well as an expanded comparison of the method in the SI.

First, we added the following text referencing smoothing as a viable method to treat data with high levels of noise
40 when discussing previous work in the Introduction (Page 4, Line 1 to 8 of the revised manuscript):

"Smoothing methods based on robust nonlinear data smoothing algorithms have been used historically to improve the signal-to-noise for data that includes occasional high signals due to random noise and other events that can bias the measurement (Beaton and Tukey, 1974; Velleman, 1977; Goring and Nikora, 2002). Smoothing algorithms separate data into a smoothed sequence that can be used to represent the baseline and a residual sequence composed of the "noise". Recently, Liu et al. (2018) used a smoothing algorithm based on a 24-hour running median to mask short-term local events with an average duration of $0.5 \pm 6$ minutes due to nearby road traffic using Condensation Particle Counter (CPC) number concentration data at Ross Island, in Antarctica during the ARM West Antarctic Radiation Experiment (Lubin et al., 2020).

Next, we added a discussion after smoothing our data at ENA in the Results (Section 4.4.3 - Comparison of ENA-AM to other masks) where we compare our mask, ENA-AM, to other methods (e.g. meteorology, metadata, other mathematical algorithms, including the coefficient of variation, etc.). The following text has been added on Page 27, Line 9-16 of the revised manuscript:

"Application of smoothing algorithms have been shown to be effective in filtering measurements affected by events lasting less than 1 hour (Liu et al., 2018) and that are associated with rapid increases in $N_{tot}$ (up to $8,520 \pm 36,780$ cm$^{-3}$) and cloud condensation nuclei concentrations $> 1,000$ cm$^{-3}$. While signals with these characteristics are present at ENA, there are also longer events that last several hours due to the complex sources associated with the local airport operations. When we applied the method at ENA, 98% of the C1 $N_{tot}$ data in the summer was masked (see SI Section SI.3 and Figure SI.4 for further information). Further optimization would be required for locations such as ENA as the method is better suited for more remote locations with less pervasive local sources, such as are encountered on a ship or remote island (Goring and Nikora, 2002)."

Lastly, to include a more complete comparison of the application of smoothing at ENA we added a new section in the SI, Section SI.5, where we include a Figure and discussion. The new section text and figure are copied below:

**"SI.5. Comparison of ENA-AM to a smoothing algorithm**

We applied a smoothing algorithm based on the one that Liu et al. (2018) developed for the AOS CPC during AWARE to our data at ENA. A 24-hour running median was used to mask the $N_{tot}$ one-minute data collected during the summer at C1. Figure SI.4 shows a comparison between ENA C1 data filtered using ENA-AM and the smoothing method. After applying the smoothing algorithm 2% of the original data were retained.

[Figure]

Figure SI.5. $N_{tot}$ data at C1 in the summer after applying a 24-hour running median smoothing algorithm (red) and ENA-AM (green)."

**(R2.2.a)** *Parameterizations of σb and alpha need to be improved. Page 7, Line 6: "We determined the standard deviation of the data below the median (σb) of Ntot for each of the two one-month periods." This sounds arbitrary and needs explanation. Is two months the result of a sensitivity check? Is it related to the time resolution of Ntot?*

**[Resp.]:** We agree that the sentence is not clear. We have clarified the choice of using one-month periods in Page 7, Lines 24-28 of the revised manuscript:

"The utilization of one-month time periods was chosen to limit biases in the characterization of the regional baseline after testing a range of periods from 2 weeks to 2 months. At ENA, we observed that when using longer periods of time (> 1 months), $\sigma_b$ removed long-term variability associated with seasonal changes. Simultaneously, considering shorter time (< 4 weeks), $\sigma_b$ were unable to retain periods when ENA was affected by episodes of long-range transport of continental air masses."

**(R2.2.b)**. *I suppose the moving medians are used here for all the data points? Why below-median, not another percentile?*

**[Resp.]:** We agree that without further explanations the choice of the median as threshold sounds arbitrary. We have clarified the explanation on Page 7, Line 29 – Page8, Line 2 of the revised manuscript:

"An alternative parameterization would be to use the standard deviation between the first and the third quartiles. This approach has been shown to be effective for masking continuous time series of greenhouse gas measurements that present daily and monthly natural fluctuations and positive short-term spikes (seconds to minutes) due to local emissions (El Yazidi et al., 2018). We tested this alternative at ENA and observed similar results for both methods. The data filtered using $\sigma_b$ agreed with the data filtered between the first and third quartiles 98.6% of the time."

**(R2.2.c)**. *The authors only tested four scenarios in Table 1. Can the authors do a sensitivity test with more data points?*

**[Resp.]:** We thank Referee #2 for this suggestion. We added more data points to our sensitivity analysis and we improved the discussion of the application and the parametrization of the algorithm in the revised manuscript.

Table 1 of the revised manuscript shows the six scenarios that have been tested:

**Table 1.** Standard deviation algorithm input parameters tested at C1 and S1 in the summer.

|  | Random Walk (RW) Threshold | Two-Point (TP) Threshold |
|---|---|---|
| $\alpha = 0.5$ | α05-RW | α05-TP |
| $\alpha = 1$ | α1-RW | α1-TP |
| $\alpha = 3$ | α3-RW | α3-TP |

We added a flow chart to describe the procedure that makes the application of the algorithm to other studies clearer in Section 3.2 (Figure 2 of the revised manuscript):

"The flow chart in Figure 2 describes the requirements and recommended procedures to apply the algorithm to data affected by local aerosol events. A perfect algorithm would identify only the noise and retain all of the natural variability. Since data may include periods when the local sources are less variable than the natural baseline and/or the baseline has more variability than the local sources, no separation will be perfect. Here, we test and develop an algorithm optimized to balance the separation of the noise from the baseline."

[Figure]

**Figure 2.** Flow chart to apply the standard deviation algorithm to high time-resolution aerosol data**.**

Next, we reorganized Section 4.5.1 "Algorithm parameterization and validation" (Section 4.4.1 of the revised manuscript) to improve the discussion about the algorithm and how the optimal parametrization was chosen to maximize the identification of local sources while at the same time minimizing loss of the natural background variability. Copied below are the new explanations that have been added to the revised manuscript, including a new Table showing the comparison of different parameterizations. The full text and figures within the reorganized Section can be read in the track changes version of the revised manuscript that has been submitted as a PDF with this response.

Page 21, Line 7-11 of the revised manuscript:

"First, we analyzed the efficiency of six parametrizations to detect high $N_{tot}$ aerosol events that were independently identified using additional collocated measurements at C1 (AOS camera and airport flight logs). Subsequently, we assessed the percentage of data removed and the $R^2$ value generated between masked $N_{tot}$ C1 and S1. Finally, we evaluated the ability of the best parametrizations to discriminate short-lived high $N_{tot}$ events from periods when ENA was affected by long-range transported continental aerosol."

Page 22, Line 19 – Page 23, Line 10 of the revised manuscript:

**Table 2.** $R^2$ values and percentage of $N_{tot}$ masked data during summer at C1 and S1 using different combinations of the α parameter and thresholding methods.

| ENA Site | | RW | | | TP | | |
|---|---|---|---|---|---|---|---|
| | | α = 0.5 | α = 1 | α = 3 | α = 0.5 | α = 1 | α = 3 |
| $R^2$ | | 0.71 | 0.76 | 0.72 | 0.88 | 0.87 | 0.79 |
| Slope ± Std. Dev. | | 0.80 ± 0.001 | 0.78 ± 0.001 | 0.75 ± 0.001 | 0.86 ± 0.001 | 0.84 ± 0.001 | 0.79 ± 0.001 |
| Data | **C1** | 19.3% | 12.5% | 5.4% | 35% | 26% | 10.6% |
| Masked | **S1** | 12.4% | 7% | 3% | 23% | 15% | 5.6% |

"Application of the RW threshold generated $R^2$ values $\leq 0.8$ between C1 and S1 independent of α. The highest α value (α = 3) with the TP threshold generated similar low correlations between the two ENA sites ($R^2 =$

0.79, slope = 0.79 ± 0.001) confirming that the α3-TP parameterization was not able to detect the all of the local aerosol events. After applying the α0.5-TP and the α1-TP parameterizations, the linear regressions and slopes were closer to unity. α0.5-TP generated a $R^2$ = 0.88 with a slope = 0.86 ± 0.001 and α1-TP a $R^2$ = 0.87 with a slope = 0.84 ± 0.001 (data fit through zero). The percentages of masked data were 35% (α0.5-TP) and 26% (α1-TP) at C1 and 23% (α0.5-TP) and 15% (α1-TP) at S1."

"Furthermore, we evaluated the ability the α0.5-TP and the α1-TP parameterization to mask short-lived high $N_{tot}$ events during periods when ENA was sampling long-range transported aerosol.

**(R2.3)** *The wind direction, wind speed, and size distribution sections are long and not necessarily related to the aerosol mask. Consider shorten them or move part of them to the supplement.?*

**[Resp.]:** We thank the reviewer for helping us streamline the text. We reorganized the text per referee recommendations. The "Wind direction and wind speed" (4.1), "Size distribution" (4.3.1) and "Variability with wind direction" (4.3.2) were moved to Supplemental Information (SI), Sections SI1, SI3 and SI4 respectively.

[revised manuscript text omitted]

---

## Author Comment (AC1)

**Manuscript No.**: acp-2020-49

**Title**: Identifying a regional aerosol baseline in the Eastern North Atlantic using collocated measurements and a mathematical algorithm to mask high submicron number concentration aerosol events

**Responses to Anonymous Referee #1**

*General Comments:*

*An algorithm is developed and presented for removing the influence of local sources from long-term, high time resolution data collected at DOE's User Facility in the Eastern North Atlantic (ENA). The technique should be useful for others interested in performing a similar data analysis. I only have minor comments which are listed below.*

**Response:** We thank Anonymous Referee #1 for his/her support of our work and the potential for it to inform analysis at other locations. We have revised the manuscript according to your suggestions. All the alterations to the manuscript are shown in the track changes revised version of the manuscript and supplemental information included within this document. Please find our point-by-point responses below.

*Detailed Comments:*

**(R1.1)** *Page 6, Lines 19 – 21: What was the RH of the sample air at C1 and S1? The size ranges of the three optical size modes should be reported at the sampling RH.*

**[Resp.]:** We thank Referee #1 for pointing out the lack of this information in our manuscript. The CPC and UHSAS sample flows, at C1 and S1, are dried using a shared Nafion dryer (Uin et al., 2019). Tests conducted at ARM AOSes showed that utilization of the Nafion dryer reduces the RH of CPC and UHSAS samples to values ≤30%. We have incorporated this information in the text on Page 5, Line 31 of the revised manuscript.

> "CPC and UHSAS sample flows are dried using a shared Naflon dryer that reduces the RH of the samples to ≤30% (Uin et al., 2019)."

**(R1.2)** *Page 9, Line 18: Change to ". . ..to create AN aerosol mask. . .".*

**[Resp.]:** The expression has been corrected as suggested (Page 10, Line 18 of the revised manuscript):

> "To understand the frequency and direction from which local aerosols originate at ENA, we present mean aerosol $N_{tot}$ and $N_{UHSAS}$ as a function of wind direction. $N_{tot}$ and $N_{UHSAS}$ are used to understand the directional and temporal influence of observed high aerosol concentrations at C1 and S1 and to evaluate the use of wind direction data to create an aerosol mask at ENA."

**(R1.3)** *Figure 2 : The axes don't seem to be labeled correctly. The x-axis should be nm, not nm^-3. Is the y-axis dN/dlogD?*

**[Resp.]:** We interpret and respond to this comment as being in reference to Figure 4 since Figure 2 is wind roses. We thank Referee #1 for catching the typos in our label axes. The label axes have been corrected to $D_p$ (nm) and $dN/dlogD_p$ ($cm^{-3}$) as shown below. Note that the figure has been moved to Supplemental Information (Figure SI.3) following the suggestion of Referee #2.

[Figure]

**(R1.4)** *Figure 7: It might be easier to see differences between C1 and S1 if one were plotted with slightly larger markers and presented behind the other one.*

**[Resp.]:** The size of the pink markers representative of $N_{tot}$ at S1 in Figure 7 (Figure 5, page 18 in the revised manuscript) have been modified as suggested. The figure from the revised manuscript is copied below.

[Figure]

**Figure 5**. One-second $N_{tot}$ during two one-day periods at C1 (orange) and S1 (pink) raw in the summer. (a) Typical day when C1 sampled higher $N_{tot}$ than S1, (b1) typical day when S1 sampled higher $N_{tot}$ than C1. To highlight the smaller peaks, the four highest peaks are off-scale bythe factors indicated in the figure.

**(R1.5)** *Page 16, Line 11: Should be "Figure 8", not 8b.*

**[Resp.]:** We thank Referee #1 for noticing this discrepancy in the text. "Figure 8b" is incorrect as the text refers to Figure 7b of the original manuscript, now Figure 5b in the revised manuscript. The text has been changed (Page 17, Line 11 of the revised manuscript):

> "Figure 5b shows a time period when the overall trend is the reverse."

**(R1.6)** *Figure 8: Wind direction should be plotted on an additional y-axis axis rather than using the vertical bars as an indication of wind direction values.*

**[Resp.]:** We thank Referee #1 for this suggestion. We have modified Figure 8 (Fig. 6 of the revised manuscript) as suggested with the wind direction plotted on an additional y-axis. See the new figure below.

[Figure]

**Figure 6**. $N_{tot}$ and wind direction at C1 (orange) and S1 (pink) on August 3, 2017. Yellow and grey periods indicate when the AOS cameras observed trucks on the runway (yellow) and planes near the terminal building (grey).

**(R1.7)** *Page 19, Line 4: Define σb.*

**[Resp.]:** We initially define "$\sigma_b$" on Page 7, Line 20 in Section 3 "Data analysis" of the original manuscript.

> "We determined the standard deviation of the data below the median ($\sigma_b$) of $N_{tot}$ for each of the two one-month periods."

For added clarity in Section 4 "Results and discussion", per Referee #1's suggestion, we have changed the text to also define "$\sigma_b$" on Page 21, Line 3-4 of the revised manuscript:

> "To apply a mathematical algorithm to mask high $N_{tot}$ events at C1 and S1, we first calculate the standard deviation of the data below the median ($\sigma_b$)".

**(R1.8)** *Page , Figure 11b: It is hard to differentiate the dark green and black markers. Maybe make the dark green a brighter color?*

**[Resp.]:** Figure 11b of the original manuscript, now Figure 10b in the revised manuscript, has been modified as suggested and is shown below.

[revised manuscript text omitted]

5 **SI.5. Comparison of ENA-AM to a smoothing algorithm**

We applied a smoothing algorithm based on the one that Liu et al. (2018) developed for the AOS CPC during AWARE to our data at ENA. A 24-hour running median was used to mask the $N_{tot}$ one-minute data collected during the summer at C1. Figure SI.5 shows a comparison between ENA C1 data filtered using ENA-AM and the smoothing method. After applying the smoothing algorithm only the 2% of the dataset was retained.

[Figure]

**Figure SI.5**. $N_{tot}$ data at C1 in the summer after applying a 24-hours running median smoothing algorithm (red) and ENA-AM (green).

**SI.6. Comparison of ENA-AM to the filter developed by Zheng et al. (2018)**

[revised manuscript text omitted]